# Norm-based Generalization Bounds for Sparse Neural Networks

**Tomer Galanti**
Center for Brains, Mind, and Machines
Massachusetts Institute of Technology
galanti@mit.edu

**Mengjia Xu**
Department of Data Science
New Jersey Institute of Technology
mx6@njit.edu

**Liane Galanti**
School of Computer Science
Tel Aviv University
lianegalanti@mail.tau.ac.il

**Tomaso Poggio**
Center for Brains, Mind, and Machines
Massachusetts Institute of Technology
tp@csail.mit.edu

## Abstract

In this paper, we derive norm-based generalization bounds for sparse ReLU neural networks, including convolutional neural networks. These bounds differ from previous ones because they consider the sparse structure of the neural network architecture and the norms of the convolutional filters, rather than the norms of the (Toeplitz) matrices associated with the convolutional layers. Theoretically, we demonstrate that these bounds are significantly tighter than standard norm-based generalization bounds. Empirically, they offer relatively tight estimations of generalization for various simple classification problems. Collectively, these findings suggest that the sparsity of the underlying target function and the model's architecture plays a crucial role in the success of deep learning.

## 1 Introduction

Over the last decade, deep learning with large neural networks has significantly advanced the solution of a myriad of tasks. These include image classification [1, 2, 3], language processing [4, 5, 6], interactions with open-ended environments [7, 8], and code synthesis [9]. Contrary to traditional theories such as [10], recent findings [11, 12] indicate that deep neural networks can generalize effectively even when their size vastly exceeds the number of training samples.

To address this question, recent work has proposed different generalization guarantees for deep neural networks based on various norms of their weight matrices [13, 14, 15, 16, 17, 18, 19, 20, 21, 22, 23]. Many efforts have been made to improve the tightness of these bounds to realistic scales. Some studies have focused on developing norm-based generalization bounds for complex network architectures, such as residual networks [24]. Other studies investigated ways to reduce the dependence of the bounds on the product of spectral norms [21, 25], or to use compression bounds based on PAC-Bayes theory [26, 27], or on the optimization procedure used to train the networks [19, 28, 29]. However, most of this research is centered around fully-connected networks, which generally underperform compared to other architectures like convolutional networks [30], residual network [1] and transformers [4, 31]. Thus, the ability of these bounds to explain the success of contemporary architectures is rather limited.

To fully understand the success of deep learning, it is necessary to analyze a wider scope of architectures beyond fully-connected networks. An interesting recent direction [32, 33] introduces generalization bounds for neural networks with shared parameters, such as convolutional neural networks. For example, [32] showed that by taking into account the structure of the convolutional

layers, we can derive generalization bounds with a norm component smaller than the norm of the associated linear transformation. However, many questions remain unanswered, including **(a)** *Why certain architectures, such as convolutional networks* [30] *and MLP-mixers* [34], *perform better than fully-connected neural networks?* **(b)** *Is weight sharing necessary for the success of convolutional neural networks?* **(c)** *Can we establish norm-based generalization bounds for convolutional neural networks that are reasonably tight in practical settings?* In this paper, we contribute to an understanding of all three questions.

## 1.1 Related Work

**Approximation guarantees for multilayer sparse networks.** While fully-connected networks, including shallow networks, are universal approximators [35, 36] of continuous functions, they are largely limited in theory and in practice. Classic results [37, 38, 39, 40, 41] show that, in the worst-case, the number of parameters required to approximate a continuously differentiable target functions (with bounded derivatives) grows exponentially with the input dimension, a property known as the "curse of dimensionality".

A recent line of work [42, 43, 44] shows that the curse of dimensionality can be avoided by deep, sparse networks, when the target function is itself compositionally sparse. Furthermore, it has been conjectured that efficiently computable functions, that is functions that are computable by a Turing machine in polynomial time, are compositionally sparse. This suggests, in turns, that, for practical functions, deep and sparse networks can avoid the curse of dimensionality. These results, however, lack any implication about generalization; in particular, they do not show that overparametrized sparse networks have good generalization.

**Norm-based generalization bounds.** A recent thread in the literature [13, 14, 15, 16, 17, 18, 19, 20, 21, 23] has introduced norm-based generalization bounds for neural networks. In particular, let $S = \{(x_i, y_i)\}_{i=1}^m$ be a training dataset of $m$ independently drawn samples from a probability measure $P$ defined on the sample space $\mathcal{X} \times \mathcal{Y}$, where $\mathcal{X} \subset \mathbb{R}^d$ and $\mathcal{Y} = \{\pm 1\}$. A fully-connected network is defined as $f_w(x) = W^L \sigma(W^{L-1} \sigma(\dots \sigma(W^2 \sigma(W^1 x)) \dots))$, where $W^l \in \mathbb{R}^{d_{l+1} \times d_l}$ and $\sigma(x)$ is the element-wise ReLU activation function $\max(0, x)$. A common approach for estimating the gap between the train and test errors of a neural network is to use the Rademacher complexity of the network. For example, in [13], an upper bound on the Rademacher complexity is introduced based on the norms of the weight matrices of the network of order $\mathcal{O}(\frac{2^L}{\sqrt{m}} \prod_{l=1}^L \|W^l\|_F)$. Later, [14] showed that the exponential dependence on the depth can be avoided by using the contraction lemma and obtained a bound that scales with $\mathcal{O}(\sqrt{L})$.

While these results provide solid upper bounds on the test error of deep neural networks, they only take into account very limited information about the architectural choices of the network. In particular, when applied to convolutional networks, the matrices $W^l$ represent the linear operation performed by a convolutional layer whose filters are $w^l$. However, since $W^l$ applies $w^l$ to several patches ($d_l$ patches), we have $\|W^l\|_F = \sqrt{d_l}\|w^l\|_F$. As a result, the bound scales with $\mathcal{O}(\sqrt{\prod_{l=1}^{L-1} d_l})$, that grows exponentially with $L$. This means that the bound is not suitable for convolutional networks with many layers as it would be very loose in practice. In this work, we establish generalization bounds that are customized for convolutional networks and scale with $\prod_{l=1}^L \|w^l\|_F$ instead of $\prod_{l=1}^L \|W^l\|_F$.

In [45] they conducted a large-scale experiment evaluating multiple norm-based generalization bounds, including those of [17, 14]. They argued that these bounds are extremely loose and negatively correlated with the test error. However, in all of these experiments, they trained the neural networks with the cross-entropy loss which implicitly maximizes the network's weight norms once the network perfectly fits the training data. This can explain the observed negative correlation between the bounds and the error. In this work, we empirically show that our bounds provide reasonably tight estimations of the generalization gap for convolutional networks trained with weight normalization and weight decay using the MSE loss.

**Generalization bounds for convolutional networks.** Several recent papers have introduced generalization bounds for convolutional networks that take into account their unique structure. In [23], they introduced a generalization bound for neural networks with weight sharing. However, this bound only holds under the assumption that the weight matrices are orthonormal, which is not real-

istic in practice. In [33], they introduced generalization bounds for convolutional networks based on parameter counting. However, this bound scales roughly as the square root of the ratio between the number of parameters and the number of samples, which is vacuous when the network is overparameterized. In [32], they extended the generalization bounds of [17] for convolutional networks where the linear transformations $W^l$ at each layer are replaced with the trainable parameters. While this paper provides generalization bounds in which each convolutional filter contributes only once to the bound, it does not hold when different filters are used for different patches, even if their norms are the same. In short, their analysis treats different patches as "datapoints" in an augmented problem where only one linear function is applied at each layer. If several choices of linear functions (different weights for different patches) are allowed, the capacity of the function class would increase. Although all of these papers offer generalization guarantees for convolutional networks, they base their findings either on the number of trainable parameters or on weight sharing. Notably, none of these studies directly address the question of whether weight sharing is essential for the effective generalization of convolutional networks. Furthermore, none provide empirical evidence to confirm that their bounds are reasonably tight in practical settings. In our previous work [46], we derived generalization bounds using a technique similar to the one employed here. The results in this paper extend those preliminary results to a more general and more detailed formulation.

## 1.2 Contributions

In this work, we study the generalization guarantees of a broad class of sparse deep neural networks [42], such as convolutional neural networks. Informally, a sparse neural network is a graph of neurons represented as a Directed Acyclic Graph (DAG), where each neuron is a function of a small set of other neurons. We show how a simple modification to the classic norm-based generalization bound of [14] yields significantly tighter bounds for sparse neural networks (e.g., convolutional networks). Unlike previous bounds [33, 32, 47], our analysis demonstrates how to obtain generalization guarantees for sparse networks, without incorporating weight sharing, while having a weak dependence on the actual size of the network. These results suggest that it is possible to obtain good generalization performance with sparse neural networks without relying on weight sharing. Finally, we conduct multiple experiments to evaluate our bounds for overparameterized convolutional neural networks trained on simple classification problems. These experiments show that in these settings, our bound is significantly tighter than many bounds in the literature [14, 33, 32, 47]. As a result, this research provides a better understanding of the pivotal influence of the structure of the network's architecture [30, 34, 2] on its test performance.

## 2 Problem Setup

We consider the problem of training a model for classification. Formally, the task is defined by a distribution $P$ over samples $(x, y) \in \mathcal{X} \times \mathcal{Y}$, where $\mathcal{X} \subset \mathbb{R}^{c_0 \times d_0}$ is the instance space (e.g., images), and $\mathcal{Y} \subset \mathbb{R}^C$ is a label space containing the $C$-dimensional one-hot encodings of the integers $1, \ldots, C$. When thinking about the samples as images, we view $c_0$ as the number of input channels and $d_0$ as the image size. We consider a hypothesis class $\mathcal{F} \subset \{f' : \mathcal{X} \to \mathbb{R}^C\}$ (e.g., a neural network architecture), where each function $f_w \in \mathcal{F}$ is specified by a vector of parameters $w \in \mathbb{R}^N$ (i.e., trainable parameters). A function $f_w \in \mathcal{F}$ assigns a prediction to an input point $x \in \mathcal{X}$, and its performance on the distribution $P$ is measured by the *expected error*, $\mathrm{err}_P(f_w) := \mathbb{E}_{(x,y) \sim P}[\mathbb{I}[\max_{j \neq y}(f_w(x_i)_j) \geq f_w(x_i)_y]]$, where $\mathbb{I} : \{\mathrm{True}, \mathrm{False}\} \to \{0, 1\}$ be the indicator function (i.e., $\mathbb{I}[\mathrm{True}] = 1$ and vice versa). Since we do not have direct access to the full population distribution $P$, the goal is to learn a predictor, $f_w$, from some training dataset $S = \{(x_i, y_i)\}_{i=1}^m$ of independent and identically distributed (i.i.d.) samples drawn from $P$ along with regularization to control $f_w$'s complexity.

### 2.1 Rademacher Complexities

We examine the generalization abilities of overparameterized neural networks by investigating their Rademacher complexity. This quantity can be used to upper bound the worst-case generalization gap (i.e., the distance between train and test errors) of functions from a certain class. It is defined as the expected performance of the class when averaged over all possible labelings of the data, where the

labels are chosen independently and uniformly at random from the set $\{\pm 1\}$. In other words, it is the average performance of the function class on random data. For more information, see [48, 49, 50].

**Definition 2.1** (Rademacher Complexity)**.** Let $\mathcal{F}$ be a set of real-valued functions $f_w : \mathcal{X} \to \mathbb{R}^C$ defined over a set $\mathcal{X}$. Given a fixed sample $X \in \mathcal{X}^m$, the empirical Rademacher complexity of $\mathcal{F}$ is defined as follows: $\mathcal{R}_X(\mathcal{F}) := \frac{1}{m}\mathbb{E}_{\xi:\xi_{ir}\sim U[\{\pm 1\}]}\left[\sup_{f_w \in \mathcal{F}}\left|\sum_{i=1}^{m}\sum_{r=1}^{C}\xi_{ir}f_w(x_i)_r\right|\right].$

In contrast to the Vapnik–Chervonenkis (VC) dimension, the Rademacher complexity has the added advantage that it can be upper bounded based on a finite sample. The Rademacher complexity can be used to upper bound the gap between test and train errors of a certain class of functions [48]. In the following lemma we bound the gap between the test error and the empirical margin error $\mathrm{err}_S^\gamma(f_w) = \frac{1}{m}\sum_{i=1}^{m}\mathbb{I}[\max_{j\neq y}(f_w(x_i)_j) + \gamma \geq f_w(x_i)_y].$

**Lemma 2.2.** *Let $P$ be a distribution over $\mathbb{R}^{c_0 d_0} \times [C]$ and $\mathcal{F} \subset \{f' : \mathcal{X} \to \mathbb{R}^C\}$. Let $S = \{(x_i, y_i)\}_{i=1}^{m}$ be a dataset of i.i.d. samples selected from $P$ and $X = \{x_i\}_{i=1}^{m}$. Then, with probability at least $1 - \delta$ over the selection of $S$, for any $f_w \in \mathcal{F}$, we have*

$$\mathrm{err}_P(f_w) - \mathrm{err}_S^\gamma(f_w) \leq \frac{2\sqrt{2}}{\gamma}\cdot\mathcal{R}_X(\mathcal{F}) + 3\sqrt{\frac{\log(2/\delta)}{2m}}. \tag{1}$$

The above bound is decomposed into two parts; one is the Rademacher complexity and the second scales as $\mathcal{O}(1/\sqrt{m})$ which is small when $m$ is large. In section 3 we derive norm-based bounds on the Rademacher complexity of sparse networks. This lemma and the rest of the mathematical statements are proven in the appendix.

## 2.2 Architectures

A neural network architecture can be formally defined using a Directed Acyclic Graph (DAG) $G = (V, E)$. The class of neural networks associated with this architecture is denoted as $\mathcal{F}_G$. The set of neurons in the network is given by $V = \bigcup_{l=0}^{L}\{z_1^l, \ldots, z_{d_l}^l\}$, which is organized into $L$ layers. An edge $(z_i^l, z_j^{l-1}) \in E$ indicates a connection between a neuron in layer $l - 1$ and a neuron in layer $l$. The full set of neurons at the layer $l$th is denoted by $v^l := (z_j^l)_{j=1}^{d_l}$.

A neural network $f_w : \mathbb{R}^{c_0 \times d_0} \to \mathbb{R}^C$ takes "flattened" images $x$ as input, where $c_0$ is the number of input channels and $d_0$ is the image dimension represented as a vector. Each neuron $z_i^l : \mathbb{R}^{c_0 \times d_0} \to \mathbb{R}^{c_l}$ computes a vector of size $c_l$ (the number of channels in layer $l$). To avoid confusion, in our definition, we think of each neuron as a vector of dimension $c_l$. This is analogous to a pixel holding three coordinates of RGB. The set of predecessor neurons of $z_i^l$, denoted by $\mathrm{pred}(l, i)$, is the set of all $j \in [d_{l-1}]$ such that $(z_i^l, z_j^{l-1}) \in E$, and $v_i^l := (z_j^l)_{j\in\mathrm{pred}(l,i)}$ denotes the set of predecessor neurons of $z_i^l$. The network is recursively defined as follows:

$$\forall r \in [C] : \ f_w(x)_r := \sum_{i=1}^{d_{L-1}}\langle w_{ri}^L, z_i^{L-1}(x)\rangle,$$

where $w_{ri}^L \in \mathbb{R}^{c_{L-1}}$, $z_i^l(x) := \sigma(w_i^l v_i^{l-1}(x))$, $w_i^l \in \mathbb{R}^{c_l \times (c_{l-1}\cdot|\mathrm{pred}(l-1,i)|)}$ is a weight matrix, $x = (z_j^0(x))_{j=1}^{d_0}$, each $z_j^0(x)$ is a vector of dimension $c_0$ representing the $j$th "pixel" of $x$ and $\sigma$ is the ReLU activation function. For simplicity, we denote $w^L := W^L = (w_{ri})_{r,i}$.

The degree of sparsity of a neural network can be measured using the degree of the graph, which is defined as the maximum number of predecessors for each neuron. Specifically, the degree of a neural network architecture $G$ is given by: $\deg(G) := \max_{l\in[L]}\deg(G)_l$, where $\deg(G)_l := \max_{j\in[d_l]}|\mathrm{pred}(l, j)|$ is the maximal degree of the $l$th layer.

**Convolutional neural networks.** A special type of compositionally sparse neural networks is convolutional neural networks. In such networks, each neuron acts upon a set of nearby neurons from the previous layer, using a kernel shared across the neurons of the same layer.

To formally analyze convolutional networks, we consider a broader set of neural network architectures that includes sparse networks with shared weights. Specifically, for an architecture $G$ with $|\mathrm{pred}(l, j)| = k_l$ for all $j \in [d_l]$, we define the set of neural networks $\mathcal{F}_G^{\mathrm{sh}}$ to consist of all neural

networks $f_w \in \mathcal{F}_G^{\text{sh}}$ that satisfy the weight sharing property $w^l := w_{j_1}^l = w_{j_2}^l$ for all $j_1, j_2 \in [d_l]$ and $l \in [L]$. Convolutional neural networks are essentially sparse neural networks with shared weights and locality (each neuron is a function of a set of nearby neurons of its preceding layer).

**Norms of neural networks.** As mentioned earlier, previous papers (e.g., [14]) proposed different generalization bounds based on different types of norms for measuring the complexity of fully-connected networks. One approach that was suggested by [14] is to use the product of the norms of the weight matrices given by $\tilde{\rho}(w) := \prod_{l=1}^L \|W^l\|_F$.

In this work, we derive generalization bounds based on the product of the maximal norms of the kernel matrices across layers, defined as: $\rho(w) := \|w^L\|_F \cdot \prod_{l=1}^{L-1} \max_{j \in [d_l]} \|w_j^l\|_F$, where $\|\cdot\|_F$ and is the Frobenius norm. For a convolutional neural network, we have a simplified form of $\rho(w) = \prod_{l=1}^L \|w^l\|_F$, due to the weight sharing property. This quantity is significantly smaller than the quantity $\tilde{\rho}(w) = \|w^L\|_F \cdot \prod_{l=1}^{L-1} \sqrt{\sum_{j=1}^{d_l} \|w_j^l\|_F^2}$ used by [14]. For instance, when weight sharing is applied, we can see that $\tilde{\rho}(w) = \rho(w) \cdot \sqrt{\prod_{l=1}^{L-1} d_l}$ which is significantly larger than $\rho(w)$.

**Classes of interest.** In the next section, we study the Rademacher complexity of classes of compositionally sparse neural networks that are bounded in norm. We focus on two classes: $\mathcal{F}_{G,\rho} := \{f_w \in \mathcal{F}_G \mid \rho(w) \le \rho\}$ and $\mathcal{F}_{G,\rho}^{\text{sh}} := \{f_w \in \mathcal{F}_G^{\text{sh}} \mid \rho(w) \le \rho\}$, where $G$ is a composition-ally sparse neural network architecture and $\rho$ is a bound on the norm of the network parameters.

## 3 Theoretical Results

In this section, we introduce our main theoretical results. The following theorem provides a bound on the Rademacher complexity of the class $\mathcal{F}_{G,\rho}$ of networks of architecture $G$ of norm $\le \rho$.

**Proposition 3.1.** *Let $G$ be a neural network architecture of depth $L$ and let $\rho > 0$. Let $X = \{x_i\}_{i=1}^m$ be a set of samples. Then,*

$$
\mathcal{R}_X(\mathcal{F}_{G,\rho}) \le \frac{\rho}{m} \cdot \left( 1 + \sqrt{2(\log(2)L + \sum_{l=1}^{L-1} \log(\deg(G)_l) + \log(C))} \right)
$$
$$
\cdot \sqrt{\max_{j_0,\ldots,j_L} \prod_{l=1}^{L-1} |\text{pred}(l, j_l)| \cdot \sum_{i=1}^m \|z_{j_0}^0(x_i)\|_2^2},
$$

*where the maximum is taken over $j_0, j_1, \ldots, j_L$, such that, $j_{l-1} \in \text{pred}(l, j_l)$ for all $l \in [L]$.*

The proof for this theorem builds upon the proof of Theorem 1 in [14]. A sketch of the proof is presented in Section 3.1. As we show next, by combining Lemma 2.2 and Proposition 3.1 we can obtain an upper bound on the test error of compositionally sparse neural networks.

**Theorem 3.2.** *Let $P$ be a distribution over $\mathbb{R}^{c_0 d_0} \times \{\pm 1\}$. Let $S = \{(x_i, y_i)\}_{i=1}^m$ be a dataset of i.i.d. samples selected from $P$. Then, with probability at least $1 - \delta$ over the selection of S, for any $f_w \in \mathcal{F}_G$,*

$$
\text{err}_P(f_w) - \text{err}_S^\gamma(f_w) \le \frac{2\sqrt{2}(\rho(w) + 1)}{\gamma m} \cdot \left( 1 + \sqrt{2(\log(2)L + \sum_{l=1}^{L-1} \log(\deg(G)_l) + \log(C))} \right)
$$
$$
\cdot \sqrt{\max_{j_0,\ldots,j_L} \prod_{l=1}^{L-1} |\text{pred}(l, j_l)| \cdot \sum_{i=1}^m \|z_{j_0}^0(x_i)\|_2^2} + 3\sqrt{\frac{\log(2(\rho(w) + 2)^2/\delta)}{2m}},
$$

*where the maximum is taken over $j_0, \ldots, j_L$, such that, $j_{l-1} \in \text{pred}(l, j_l)$ for all $l \in [L]$.*

The theorem above provides a generalization bound for neural networks of a given architecture $G$. To understand this bound, we first analyze the term $\Delta := \max_{j_0,\ldots,j_L} \prod_{l=1}^{L-1} |\text{pred}(l, j_l)| \cdot \sum_{i=1}^m \|z_{j_0}^0(x_i)\|_2^2$. We consider a setting where $d_0 = 2^L$, $c_l = 1$ and each neuron takes two neurons as input, $k_l := |\text{pred}(l, j)| = 2$ for all $l \in [L]$ and $j \in [d_l]$. In particular, $\prod_{l=1}^{L-1} k_l = 2^{L-1}$ and $z_j^0(x_i)$ is the $j$th pixel of $x_i$. Therefore, we have $\Delta = \frac{d_0}{2} \cdot \max_{j_0} \sum_{i=1}^m \|z_{j_0}^0(x_i)\|_2^2$. We

note that in the worst-case, when all of the norm $\|x_i\|$ is concentrated in the first pixel $z_1^0(x_i)$, we have $\Delta = \frac{d_0}{2} \cdot \sum_{i=1}^m \|x_i\|^2$. But in practice, the norms of the pixels $z_j^0(x_i)$ are typically more evenly distributed. For instance, we can assume that the pixels are $\beta$-balanced. Meaning, $\forall i \in [m]:\ \max_{j \in [d_0]} \|z_j^0(x_i)\|^2 \le \beta \operatorname{Avg}_{j \in [d_0]}[\|z_j^0(x_i)\|^2] = \frac{\beta}{d_0} \|x_i\|^2$ (for some constant $\beta > 0$). Note that the rate $\beta$ is a property exclusively dependent on the data as it is independent of the architecture and the training process. In particular, we obtain that $\Delta \le \frac{\beta}{2} \sum_{i=1}^m \|x_i\|^2$. In addition, we note that the second term in the bound is typically smaller than the first term as it scales with $\sqrt{\log(\rho(w))}$ instead of $\rho(w)$ and has no dependence on the size of the network. Therefore, in this case, our bound can be simplified to $\mathcal{O}(\frac{\rho(w)}{\sqrt{m}} \sqrt{L\beta \operatorname{Avg}_{i=1}^m[\|x_i\|^2]})$.

**Bounds for convolutional networks.** As previously stated in section 2, convolutional neural networks utilize weight sharing across neurons in each layer, with each neuron in the $l$th layer having $k_l$ input neurons (each of dimension $c_{l-1}$). The norm of the network is calculated as $\rho(w) = \prod_{l=1}^L \|w^l\|_F$, and the degree at each layer is simply the kernel size $\deg(G)_l = k_l$. This results in a simplified version of the theorem.

**Corollary 3.3** (Rademacher complexity of convolutional networks). *Let $G$ be a neural network architecture of depth $L$ and let $\rho > 0$. Let $X = \{x_i\}_{i=1}^m$ be a set of samples. Then,*

$$
\mathcal{R}_X(\mathcal{F}_{G,\rho}^{\mathrm{sh}}) \le \frac{\rho}{m} \cdot \left(1 + \sqrt{2(\log(2)L + \sum_{l=1}^{L-1} \log(k_l) + \log(C))}\right) \cdot \sqrt{\prod_{l=1}^{L-1} k_l \cdot \max_{j \in [d_0]} \sum_{i=1}^m \|z_j^0(x_i)\|^2},
$$

*where $k_l$ denotes the kernel size in the $l$'th layer.*

**Comparison with the bound of [14].** The result in Corollary 3.3 is a refined version of the analysis in [14] for the specific case of convolutional networks. Theorem 1 in [14] can of course be applied to convolutional networks by treating their convolutional layers as fully-connected layers. However, this approach yields a substantially worse bound compared to the one proposed in Corollary 3.3.

Consider a convolutional neural network $G$. The $l$th convolutional layer takes the concatenation of $(\sigma(z_1^l), \ldots, \sigma(z_{d_l}^l))$ as input and returns $(z_1^{l+1}, \ldots, z_{d_{l+1}}^{l+1})$ as its output. Each $z_j^{l+1}$ is computed as follows $z_j^{l+1} = w^{l+1}\sigma(v_j^l(x))$. Therefore, the matrix $W^{l+1}$ associated with the convolutional layer contains $d_{l+1}$ copies of $w^{l+1}$ and its Frobenius norm is therefore $\sqrt{d_{l+1}} \cdot \|w^{l+1}\|_F$. In particular, by applying Theorem 1 in [14], we obtain a bound that scales as $\mathcal{O}\left(\frac{\rho}{m}\sqrt{L\prod_{l=1}^{L-1} d_l \cdot \sum_{i=1}^m \|x_i\|^2}\right)$. On the other hand, we have $\prod_{l=1}^{L-1} k_l \le \prod_{l=1}^{L-1} d_l$ and $\|z_j^0(x_i)\| \le \|x_i\|$. Therefore, our bound is always smaller than $\mathcal{O}\left(\frac{\rho}{m}\sqrt{L\prod_{l=1}^{L-1} d_l \cdot \sum_{i=1}^m \|x_i\|^2}\right)$, which is the bound we obtained with [14]. In particular, if each convolutional layer has $k_l = 2$ with no overlaps and $d_0 = 2^L$, then, $d_l = 2^{L-l}$ and the bound of [14] would scale as $\mathcal{O}\left(\frac{\rho}{\sqrt{m}}\sqrt{L2^{0.5L(L-1)} \cdot \operatorname{Avg}_{i=1}^m[\|x_i\|^2]}\right)$. On the other hand, in our bound we have the term $\prod_{l=1}^{L-1} k_l = 2^{L-1}$, and therefore, our bound scales as $\mathcal{O}\left(\frac{\rho}{\sqrt{m}}\sqrt{L2^L \operatorname{Avg}_{i=1}^m[\|x_i\|^2]}\right)$ which is significantly smaller. Finally, as we discussed earlier, if the norms of the pixels of each sample $x$ are $\beta$-balanced (for some constant $\beta > 0$), our bound scales as $\mathcal{O}\left(\frac{\rho}{\sqrt{m}}\sqrt{L \operatorname{Avg}_{i=1}^m[\|x_i\|^2]}\right)$ which is smaller by a factor of $2^{0.25L(L-1)}$ than the bound of [14].

**Comparison with the bound of [33].** A recent paper [33] introduced generalization bounds for convolutional networks based on parameter counting. This bound roughly scales like $\mathcal{O}\left(\sqrt{\frac{N(\sum_{l=1}^L \|w^l\|_2 + \log(1/\gamma)) + \log(1/\delta)}{m}}\right)$, where $\gamma$ is a margin (typically smaller than 1), and $N$ is the number of trainable parameters (taking weight sharing into account by counting each parameter of convolutional filters only once). While these bounds provide improved generalization guarantees when reusing parameters, it scales as $\Omega(\sqrt{N/m})$ which is very large in practice. For example, the standard ResNet-50 architecture has approximately $N = 23M$ trainable parameters while the MNIST dataset has only $m = 50000$ training samples.

**Comparison with the bounds of [32] and [47].** Recent papers [32] introduced generalization bound for convolutional networks based on covering numbers and weight sharing. For example,

the bounds in Theorem 17 of [32] roughly scale as $\mathcal{O}\left(\frac{\prod\limits_{l=1}^{L} \|W^l\|_2}{\sqrt{m}} E(w)^{1/\alpha} \cdot I_\alpha\right)$, where $E(w) =$

$\left(\sum_{l=1}^{L-1} \frac{k_l^{\frac{\alpha}{2}} \|(w^l - u^l)^\top\|_{2,1}^\alpha}{\|w^l\|_2^\alpha} + \frac{\|w^L\|_2^\alpha}{\max\limits_i \|w^L_{i,:}\|_2^\alpha}\right)$, $k_l$ is the kernel size of the $l$th layer and $W^l$ is the matrix corresponding to the linear operator associated with the $l$th convolutional layer, $w_{i,:}$ is the $i$th row of a matrix $w$, $\alpha$ is either 2 or 2/3, $I_\alpha = L$ if $\alpha = 2$ and $I_\alpha = 1$ o.w. and $u^l$ are "reference" matrices of the same dimensions as $w^l$.

In general, neither our bounds nor those in [32] and [47] are inherently superior; with each being better in different cases. The main difference between their bounds and our bound, is that while their bounds include both multiplicative complexity term and additive complexity term, our bound includes only a multiplcative complexity term. For instance, the bound in Theorem 17 in [32] features both $\prod_{l=1}^{L} \|w^l\|_2$ and $E(w)$, whereas our bound exclusively contains the multiplicative term $\rho(w) = \prod_{l=1}^{L} \|w^l\|_F$. For certain cases, this works in favor of our bound, but at the same time, our term $\prod_{l=1}^{L} \|w^l\|_F$ is comparably larger than $\prod_{l=1}^{L} \|w^l\|_2$ due to the smaller norms used in the former. As an example of a case where our bound is superior, consider the case described after Theorem 3.2, where each convolutional layer operates on non-overlapping patches of size 2 and the channel dimension is 1 at each layer. We choose $u^l = 0$ for all $l \in [L-1]$ (which is a standard choice of reference matrices). We notice that $\|W^l\|_2 = \|w^l\|_2 = \|w^l\|_F$ since $W^l$ is a block matrix and $w^l$ is a vector. In addition, for any matrix $A$, we have $\text{rank}(A) \geq \frac{\|A^\top\|_{2,1}}{\|A\|_2} \geq \frac{\|A\|_F}{\|A\|_2} \geq 1$ and $\text{rank}(A) \geq \frac{\|A\|_2}{\max_i \|A_{i,:}\|_2} \geq 1$ (see [17]). Therefore, the bound in [32] scales as at least $\frac{\prod_{l=1}^{L} \|w^l\|_2}{\sqrt{m}} \cdot L^{3/2}$, while our bound scales as $\frac{\prod_{l=1}^{L} \|w^l\|_2}{\sqrt{m}} \sqrt{L}$ which is smaller by a factor of $L$.

**Vacuous bounds?** A uniform convergence bound for a class $\mathcal{F}$ is an upper bound on the generalization gap that uniformly holds for all $f \in \mathcal{F}$, i.e., $\sup_{f \in \mathcal{F}} |\text{err}_P(f) - \text{err}_S(f)| \leq \epsilon(m, \mathcal{F})$ (typically tends to 0 as $m \to \infty$). The Rademacher complexity bound in Lemma 2.2 is a form of uniform convergence bound. The issue with these bounds is that in interpolation regimes, where there exists a function $f \in \mathcal{F}_G$ that fits any labeling of the samples $\{x_i\}_{i=1}^m$, uniform convergence bounds are provably vacuous.

While the derivation of the bound in Theorem 3.2 follows the application of Rademacher complexities, we emphasize that it is not a uniform convergence bound and is not necessarily vacuous. Throughout the proof, we sliced the class $\mathcal{F}_G$ into subsets $\mathcal{F}_{G,\rho} = \{f \in \mathcal{F}_G \mid \rho(w) \leq \rho\}$ (for $\rho \in \mathbb{N}$) and applied Lemma 2.2 for each of these subsets. This approach yields a bound that is proportional to $\mathcal{O}(\rho/\sqrt{m})$ for each of the slices $\mathcal{F}_{G,\rho}$. We then apply a union bound to combine all of them to obtain a bound that scales as $\mathcal{O}(\rho(w)/\sqrt{m})$. This does not give a uniform convergence bound across all members of $\mathcal{F}_G$, since the bound is individualized for each member $f_w \in \mathcal{F}_G$ based on the norm $\rho(w)$. For example, for $w = 0$, the bound will be 0 which is non-vacuous.

When the learning algorithm minimizes $\rho(w)$ and the minimal norm required to fit the training labels is small, a tight bound can be achieved with a network that perfectly fits the training data. For example, suppose we have a dataset $S = \{(x_i, y_i)\}_{i=1}^m$, a target function $y(x) = \langle w^*, x \rangle$, and a hypothesis class $\mathcal{F} = \{\langle w, x \rangle \mid w \in \mathbb{R}^d\}$. A classic VC-theory bound scales as $\mathcal{O}(\sqrt{d/m})$, which is vacuous when $d \gg m$. However, a norm-based bound scales as $\mathcal{O}(\|w\|/\sqrt{m})$, which is non-vacuous for any $\|w\| \leq \|w^*\|$ (as long as $m > \|w^*\|^2$). In addition, the function $y$ can be realized by $\{\langle w, x \rangle \mid \|w\| \leq \|w^*\|\} \subset \mathcal{F}$. Specifically, for smaller $\|w^*\|$, we need fewer samples to ensure that the bound is non-vacuous for a minimal norm model that perfectly fits the training data.

### 3.1 Proof Sketch

We propose an extension to a well-established method for bounding the Rademacher complexity of norm-bounded deep networks. This approach, originally developed by [13], utilizes a "peeling" argument, where the complexity bound for a depth $L$ network is reduced to a complexity bound for a depth $L - 1$ network and applied repeatedly. Specifically, the $l$th step bounds the complexity bound for depth $l$ by using the product of the complexity bound for depth $l - 1$ and the norm of the $l$th layer. By the end of this process, we obtain a bound that depends on the term $\mathbb{E}_\xi g(|\sum_{i=1}^m \xi_i x_i|)$

($g(x) = x$ in [13] and $g = \exp$ in [14]), which can be further bounded using $\max_{x \in X} \|x\|^2$. The final bound scales with $\tilde{\rho}(w) \cdot \max_{x \in X} \|x\|$. Our extension further improves the tightness of these bounds by incorporating additional information about the network's sparsity.

To bound $\mathcal{R}_X(\mathcal{F}_{G,\rho})$ using $\rho(w)$, we notice that each neuron operates on a small subset of the neurons from the previous layer. Therefore, we can bound the contribution of a certain constituent function $z_j^l(x) = w_j^l v_j^{l-1}(x)$ in the network using the norm $\|w_j^l\|_F$ and the complexity of $v_j^{l-1}(x)$ instead of the full layer $v^{l-1}(x)$. To explain this process, we provide a proof sketch of Proposition 3.1 for convolutional networks $G = (V, E)$ with non-overlapping patches. For simplicity, we assume that $d_0 = 2^L$, $c_l = 1$, and the strides and kernel sizes at each layer are $k = 2$. In particular, the network $f_w$ can be represented as a binary tree, where the output neuron is computed as $f_w(x) = z_{j_0}^L(x) = w^L \cdot \sigma(z_1^{L-1}(x), z_2^{L-1}(x))$, $z_1^{L-1}(x) = w^{L-1} \cdot \sigma(z_1^{L-2}(x), z_2^{L-2}(x))$ and $z_2^{L-1}(x) = w^{L-1} \cdot \sigma(z_3^{L-2}(x), z_4^{L-2}(x))$ and so on. Similar to [14], we first bound the Rademacher complexity using Jensen's inequality,

$$m\mathcal{R}_X(\mathcal{F}_{G,\rho}) = \tfrac{1}{\lambda}\log\exp\left(\lambda\mathbb{E}_\xi \sup_{f_w}\sum_{i=1}^m \xi_i f_w(x_i)\right) \leq \tfrac{1}{\lambda}\log\left(\mathbb{E}_\xi \sup_{f_w}\exp\left(\left|\lambda\sum_{i=1}^m \xi_i f_w(x_i)\right|\right)\right), \quad (2)$$

where $\lambda > 0$ is an arbitrary parameter. As a next step, we rewrite right-hand side as follows:

$$\mathbb{E}_\xi \sup_{f_w}\exp\left(\lambda\left|\sum_{i=1}^m \xi_i \cdot f_w(x_i)\right|\right) = \mathbb{E}_\xi \sup_{f_w}\exp\left(\lambda\sqrt{\left|\sum_{i=1}^m \xi_i \cdot w^L \cdot \sigma(z_1^{L-1}(x_i), z_2^{L-1}(x_i))\right|^2}\right)$$

$$\leq \mathbb{E}_\xi \sup_{f_w}\exp\left(\lambda\sqrt{\|w^L\|_F^2 \cdot \sum_{j=1}^2 \left\|\sum_{i=1}^m \xi_i \cdot \sigma(z_j^{L-1}(x_i))\right\|_2^2}\right). \quad (3)$$

We notice that each $z_j^{L-1}(x)$ is itself a depth $L-1$ binary-tree neural network. Therefore, intuitively we would like to apply the same argument $L-1$ more times. However, in contrast to the above, the networks $\sigma(z_1^{L-1}(x)) = \sigma(w^{L-1}(z_1^{L-2}(x), z_2^{L-2}(x)))$ and $\sigma(z_2^{L-1}(x)) = \sigma(w^{L-1}(z_3^{L-2}(x), z_4^{L-2}(x)))$ end with a ReLU activation. To address this issue, [13, 14] proposed a "peeling process" based on Equation 4.20 in [51] that can be used to bound terms of the form $\mathbb{E}_\xi \sup_{f' \in \mathcal{F}', W: \|W\|_F \leq R}\exp\left(\alpha\left\|\sum_{i=1}^m \xi_i \cdot \sigma(Wf'(x_i))\right\|\right)$. However, this bound is not directly applicable when there is a sum inside the square root, as in equation 3 which includes a sum over $j = 1, 2$. Therefore, a modified peeling lemma is required to deal with this case.

**Lemma 3.4** (Peeling Lemma). *Let $\sigma$ be a 1-Lipschitz, positive-homogeneous activation function which is applied element-wise (such as the ReLU). Then for any class of vector-valued functions $\mathcal{F} \subset \{f = (f_1, \ldots, f_q) \mid \forall j \in [q]: f_j : \mathbb{R}^d \to \mathbb{R}^p\}$, and any convex and monotonically increasing function $g : \mathbb{R} \to [0, \infty)$,*

$$\mathbb{E}_\xi \sup_{\substack{f \in \mathcal{F} \\ W_j: \|W_j\|_F \leq R}} g\left(\sqrt{\sum_{j=1}^q \left\|\sum_{i=1}^m \xi_i \cdot \sigma(W_j f_j(x_i))\right\|_2^2}\right) \leq 2\mathbb{E}_\xi \sup_{j \in [q], f \in \mathcal{F}} g\left(\sqrt{q}R\left\|\sum_{i=1}^m \xi_i \cdot f_j(x_i)\right\|_2\right).$$

By applying this lemma $L-1$ times with $g = \exp$ and $f$ representing the neurons preceding a certain neuron at a certain layer, we can bound the term in equation 3 as follows:

$$\leq 2^L \mathbb{E}_\xi \sup_{j,w}\exp\left(\lambda\sqrt{\prod_{l=1}^L \|w^l\|_F^2 \cdot 2^L \left|\sum_{i=1}^m \xi_i x_{ij}\right|^2}\right)$$

$$\leq 2^L \sum_{j=1}^d \mathbb{E}_\xi \exp\left(\lambda 2^{L/2}\rho \cdot \left|\sum_{i=1}^m \xi_i x_{ij}\right|\right) \leq 4^L \sup_j \exp\left(\frac{\lambda^2 2^L \rho^2 \cdot \sum_{i=1}^m x_{ij}^2}{2} + \lambda 2^{L/2}\rho \cdot \sqrt{\sum_{i=1}^m x_{ij}^2}\right),$$

where the last inequality follows from standard concentration bounds (see the proof for details). Finally, by equation 2 and properly adjusting $\lambda$, we can finally bound $\mathcal{R}_X(\mathcal{F}_{G,\rho})$.

## 4 Experiments

In this section, we empirically evaluate the generalization bounds derived in section 3. In each experiment, we compare our bound with alternative bounds from the literature. We focus on simple

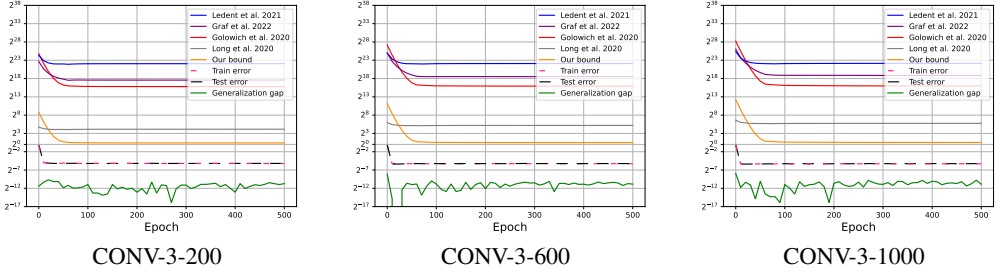

| CONV-3-200 | CONV-3-600 | CONV-3-1000 |

Figure 1: **Comparing our bound with prior bounds in the literature during training.** We plot our bound, the train and test errors, the generalization gap, and prior bounds from the literature during training. For each plot, we train a CONV-$L$-$H$ network on MNIST with a different number of channels $H$.

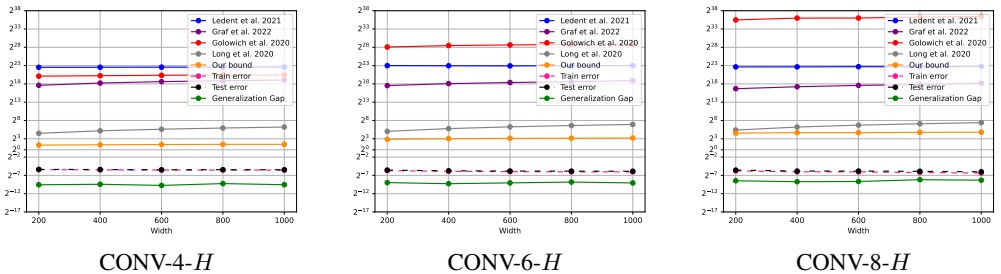

| CONV-4-$H$ | CONV-6-$H$ | CONV-8-$H$ |

Figure 2: **Varying the number of channels.** We plot our bound, the train and test errors, the generalization gap, and prior bounds from the literature at the end of training. For each plot, we train a CONV-$L$-$H$ network on MNIST with a different number of layers $L$ and channels $H$.

convolutional neural networks trained on MNIST and investigate the behavior of the bound when varying different hyperparameters. Each experiment was averaged across five runs. For additional experimental details, please refer to the appendix.

**Network architecture.** We used convolutional networks with $L$ layers and $H$ channels per layer denoted by CONV-$L$-$H$. The networks consist of a stack of $L$ $2\times 2$ convolutional layers with a stride of 1, 0 padding, and $H$ output channels, utilizing ReLU activations followed by a fully-connected layer. The overall number of trainable parameters is at least $4H + 4(L-1)H^2$.

**Optimization process.** Each model was trained using SGD for MSE-loss minimization between the logits of the network and the one-hot encodings of the training labels. We applied weight normalization [52] to all trainable layers, except for the last one, which is left un-normalized. In order to regularize the weight parameters, we used weight decay for each one of the layers of the network with the same regularization parameter $\lambda > 0$. To train each model, we used an initial learning rate of $\mu = 0.01$ that is decayed by a factor of $0.1$ at epochs 60, 100, 300, batch size 32, momentum of 0.9, and $\lambda = 3\mathrm{e}{-3}$ by default.

**Experiments.** We conducted several experiments to compare our bound to alternative bounds from the literature, when applied for neural networks trained on the MNIST dataset for classification. Throughout these experiments, we compared our bound to the one in Theorem 1 of [14], the third inequality in Theorem 2.1 of [33], Theorem 16 of [32], and Theorem 3.5 in [47] (explicitly as mentioned in their Table 3). To compute the bound in [14] for multi-class classification, we adopted a modified version based on the technique we utilized in the proof of Proposition 3.1, which allows us to extend these bounds for multi-class classification. Since [33] did not provide an explicit value for their coefficient $C$, we assumed it to be $1$. In the first experiment we trained three models of different widths for MNIST classification. As can be seen in Figure 1, our bound is significantly smaller than the alternative bounds and is surprisingly close to 1, indicating its tightness. In the second experiment, we compared our bound with alternative bounds from previous literature, focusing on convolutional neural networks with varying depths and widths. As can be seen in Figure 2, even as

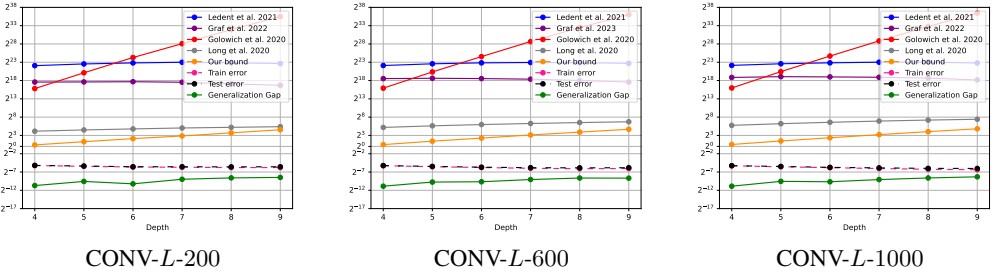

| | | |
|---|---|---|
| CONV-$L$-200 | CONV-$L$-600 | CONV-$L$-1000 |

Figure 3: **Varying the number of layers.** We plot our bound, the train and test errors, the generalization gap, and prior bounds from the literature at the end of training. For each plot, we train a CONV-$L$-$H$ network on MNIST with a different number of layers $L$ and channels $H$.

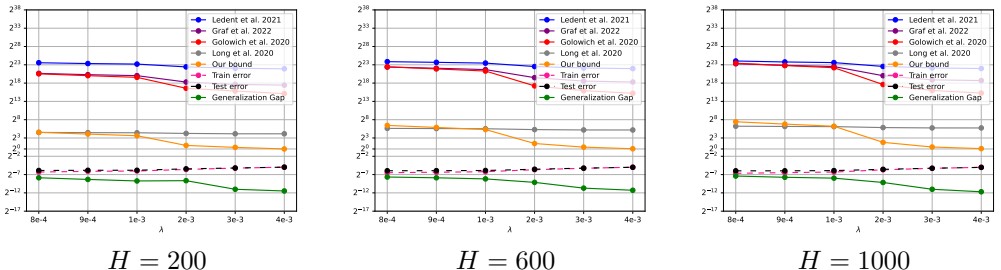

| | | |
|---|---|---|
| $H = 200$ | $H = 600$ | $H = 1000$ |

Figure 4: **Varying the regularization coefficient $\lambda$.** We plot our bound, the train and test errors, the generalization gap, and prior bounds from the literature at the end of 500 epochs. For each plot, we trained a CONV-3-$H$ network on MNIST with a varying number of channels $H$.

networks become highly overparameterized at large widths, the width does not appear to influence the results of any of the bounds. In Figure 3, we showcase the results for convolutional networks of different depths. Our bound increases exponentially with depth but rises more slowly than the bound from [14]. As an ablation study, in in Figure 4, we compared our bound with alternative bounds, this time varying the regularization coefficient $\lambda$. It is evident that our bound and some alternative bounds decrease concurrently with the generalization gap of the neural network, as desired outcome.

## 5  Conclusions

We studied the question of why certain deep learning architectures, such as convolutional networks and MLP-mixers, perform better than others on real-world datasets. To tackle this question, we derived Rademacher complexity generalization bounds for sparse neural networks, which are orders of magnitude better than a naive application of standard norm-based generalization bounds for fully-connected networks. In contrast to previous papers [33, 32], our results do not rely on parameter sharing between filters, suggesting that the sparsity of the neural networks is the critical component to their success. This sheds new light on the central question of why certain architectures perform so well and suggests that sparsity may be a key factor in their success. Even though our bounds are not practical in general, our experiments show that they are quite tight for simple classification problems, unlike other bounds based on parameter counting and norm-based bounds for fully-connected networks.

## Acknowledgements

We thank Akshay Rangamani, Lorenzo Rosasco, Brian Cheung and Eran Malach for many relevant discussions. This material is based on the work supported by the Center for Minds, Brains and Machines (CBMM), funded by NSF STC award CCF-1231216. This work has also been sponsored by DOE SEA-CROGS project (DE-SC0023191).

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

# 6 Limitations

Our work demonstrates how a simple modification to the proof in [14] can yield tighter generalization bounds for convolutional networks. Empirically, we have observed that our bound is reasonably tight in basic learning settings (e.g., training 3-layer neural networks on MNIST), which is promising. However, this bound is anticipated to be generally loose for more complex learning settings. One drawback of our bound is its dependence on depth. For instance, our factor $\rho(w)$ represents a product of $L$ weight norms, which could increase as the depth $L$ grows. Additionally, our bound contains a product $\prod_{l=1}^{L-1} k_l$, which increases exponentially with depth. As demonstrated in section 3, this indicates that the bound is likely reasonable only when $L$ depends logarithmically on the input's dimension. Consequently, our bound may not be suitable for very deep convolutional networks. Finally, building upon the analysis in [14], our bounds are specific to a certain class of ReLU neural networks and do not readily extend to architectures with different activation functions.

# 7 Broader Impact

This paper delves into the theoretical aspects of deep learning, specifically examining the reasons behind the strong generalization capabilities of sparse neural networks. Such insights may enrich our understanding of the success of deep learning and specific architectures, such as convolutional networks [30] and MLP-mixers [34]. However, since this is a theoretical study, we do not anticipate any direct negative societal impacts stemming from our research.

# 8 Additional Experimental Details

**Compute.** Each of the runs was done using a single GPU for at most 20 hours on a computing cluster with several available GPU types (e.g., GeForce RTX 2080, GeForce RTX 2080 Ti, Quadro RTX 6000, Tesla V-100, GeForce RTX A6000, A100, and GeForce GTX 1080 Ti).

**Calculating our bound.** Before discussing the calculation of the bound, let us first define how we model a convolutional neural network within the framework outlined in section 2.

We focus on ReLU convolutional networks which comprise a fully-connected layer on top of a series of convolutional layers. The $l$th convolutional layer possesses a kernel size of $k_l$. Each neuron in this $l$th layer accepts $k_l$ neurons from its predecessor layer as input (therefore its degree is $\deg(G)_l = k_l$), each with dimension $c_{l-1}$, and outputs a single output node of dimension $c_l$. The top linear layer $L$ following the series of convolutional layers receives a set of $k_L$ nodes as input (this represents the image resultant from applying multiple convolutional layers), where each node is of the output channel dimension of the previous layer. The output of this layer is of dimension $k$.

The bound derived in Theorem 3.2 of the main text can be written as follows:

$$\frac{2\sqrt{2}(\rho(w)+1)}{m} \cdot \left(1 + \sqrt{2\left(\log(2)L + \sum_{l=1}^{L-1} \log(\deg(G)_l) + \log(C)\right)}\right)$$

$$\cdot \sqrt{\max_{j_0,\ldots,j_L} \prod_{l=1}^{L-1} |\mathrm{pred}(l, j_l)| \cdot \sum_{i=1}^{m} \|z_{j_0}^0(x_i)\|_2^2} + 3\sqrt{\frac{\log(2(\rho(w)+2)^2/\delta)}{2m}},$$

To compute this bound, we set $\delta = 0.001$ (the bound holds uniformly with probability $1 - \delta$) and $\gamma = 1$. The norm $\rho(w)$ is calculated as the product of the norms of each one of the network's layers $\|w^l\|_F$. The norm of each of these layers is simply the Frobenius norm of the kernel tensor of the layer. For the last fully-connected layer we calculate the norm of the full weight matrix. For the $l$th convolutional layer, $k_l = |\mathrm{pred}(l, j_l)| = \deg(G)_l$ is the kernel's size. For instance, if the layer is a $3 \times 3$ or $2 \times 2$ convolutional layer, the size is 9 or 4, respectively. It is worth noting that the bound is independent of the size of the image $k_L$ that the fully-connected layer receives as input.

Finally, to calculate $\max_{j_1,\ldots,j_L} \sum_{i=1}^{m} \|z_{j_0}^0(x_i)\|_2^2$, with $j_{l-1} \in \mathrm{pred}(l, j_l)$. It is important to note that this term is bounded by $\max_{j_0} \sum_{i=1}^{m} \|z_{j_0}^0(x_i)\|_2^2$, where the maximum is taken over all input patches. To calculate this quantity, we sum the squared norms of the $j_0$th pixel across all samples and return the maximum value across all possible $j_0$ choices.

## 9 Proofs

**Lemma 2.2.** *Let $P$ be a distribution over $\mathbb{R}^{c_0 d_0} \times [C]$ and $\mathcal{F} \subset \{f' : \mathcal{X} \to \mathbb{R}^C\}$. Let $S = \{(x_i, y_i)\}_{i=1}^m$ be a dataset of i.i.d. samples selected from $P$ and $X = \{x_i\}_{i=1}^m$. Then, with probability at least $1 - \delta$ over the selection of $S$, for any $f_w \in \mathcal{F}$, we have*

$$\mathrm{err}_P(f_w) - \mathrm{err}_S^\gamma(f_w) \;\leq\; \frac{2\sqrt{2}}{\gamma} \cdot \mathcal{R}_X(\mathcal{F}) + 3\sqrt{\frac{\log(2/\delta)}{2m}}. \tag{1}$$

*Proof.* By Lemma 3.1 in [17], with probability at least $1 - \delta$ over the selection of $S$, for any $f_w \in \mathcal{F}$, we have

$$\mathrm{err}_P(f_w) - \mathrm{err}_S^\gamma(f_w) \;\leq\; 2\mathcal{R}_X(\mathcal{F}_{|\gamma}) + 3\sqrt{\frac{\log(2/\delta)}{2m}},$$

where $\mathcal{F}_{|\gamma} = \{(x,y) \mapsto \ell_\gamma(-\mathcal{M}(f_w(x), y)) \mid f_w \in \mathcal{F}\}$, $\ell_\gamma(u, y) := \min(1, \max(0, 1 + \frac{\mathcal{M}(u,y)}{\gamma}))$ is the ramp loss function and $\mathcal{M}(u, j) = u_j - \max_{i \neq j} u_i$ (these quantities are standard [53, 54, 55]). Next, by Lemma A.3 in [17], $\ell_\gamma(\mathcal{M}(\cdot, r))$ is a $(2/\gamma)$-Lipschitz function. Hence, by Corollary 1 in [56],

$$\mathcal{R}_X(\mathcal{F}_{|\gamma}) \;\leq\; \frac{2\sqrt{2}}{\gamma m} \cdot \mathbb{E}_{\xi_{ir} \sim U[\pm 1]} \left[ \sup_{f_w} \sum_{i,r} \xi_{ir} f_w(x_i)_r \right] \;=\; \frac{2\sqrt{2}}{\gamma} \cdot \mathcal{R}_X(\mathcal{F})$$

which completes the proof. $\qquad\square$

**Lemma 3.4** (Peeling Lemma). *Let $\sigma$ be a 1-Lipschitz, positive-homogeneous activation function which is applied element-wise (such as the ReLU). Then for any class of vector-valued functions $\mathcal{F} \subset \{f = (f_1, \ldots, f_q) \mid \forall j \in [q]: f_j : \mathbb{R}^d \to \mathbb{R}^p\}$, and any convex and monotonically increasing function $g : \mathbb{R} \to [0, \infty)$,*

$$\mathbb{E}_\xi \sup_{\substack{f \in \mathcal{F} \\ W_j:\, \|W_j\|_F \leq R}} g\left( \sqrt{\sum_{j=1}^q \left\| \sum_{i=1}^m \xi_i \cdot \sigma(W_j f_j(x_i)) \right\|_2^2} \right) \;\leq\; 2\mathbb{E}_\xi \sup_{j \in [q],\, f \in \mathcal{F}} g\left( \sqrt{q} R \left\| \sum_{i=1}^m \xi_i \cdot f_j(x_i) \right\|_2 \right).$$

*Proof.* Let $W \in \mathbb{R}^{h \times p}$ be a matrix and let $w_1, \ldots, w_h$ be the rows of the matrix $W$. Define a function $Q_j(v) := \left( \sum_{i=1}^m \xi_i \cdot \sigma(\frac{v^\top}{\|v\|_2} f_j(x_i)) \right)^2$ taking a vector $v \in \mathbb{R}^p$. We notice that

$$\sum_{j=1}^q \left\| \sum_{i=1}^m \xi_i \cdot \sigma(W_j f_j(x_i)) \right\|_2^2 \;=\; \sum_{j=1}^q \sum_{r=1}^h \|w_{jr}\|_2^2 \left( \sum_{i=1}^m \xi_i \cdot \sigma(\frac{w_{jr}^\top}{\|w_{jr}\|_2} f_j(x_i)) \right)^2$$

$$=\; \sum_{j=1}^q \sum_{r=1}^h \|w_{jr}\|_2^2 \cdot Q_j(w_{jr}).$$

For any $w_{j1}, \ldots, w_{jh}$, we have

$$\sum_{r=1}^h \|w_{jr}\|_2^2 \cdot Q_j(w_{jr}) \;\leq\; R \cdot \max_r Q_j(w_{jr}), \tag{4}$$

which is obtained for $\hat{w}_{j1}, \ldots, \hat{w}_{jh}$, where $\hat{w}_{ji} = 0$ for all $i \neq r^*$ and $\hat{w}_{jr^*}$ of norm $R$ for some $r^* \in [h]$. Together with the fact that $g$ is a monotonically increasing function, we obtain

$$\mathbb{E}_\xi \sup_{\substack{f \in \mathcal{F} \\ W_j: \|W_j\|_F \leq R}} g\left(\sqrt{\sum_{j=1}^q \left\|\sum_{i=1}^m \xi_i \cdot \sigma(W_j f_j(x_i))\right\|_2^2}\right)$$

$$\leq \mathbb{E}_\xi \sup_{\substack{f \in \mathcal{F} \\ w_1 \ldots, w_q: \|w_j\|_2 = R}} g\left(\sqrt{\sum_{j=1}^q |\sum_{i=1}^m \xi_i \cdot \sigma(w_j^\top f_j(x_i))|^2}\right)$$

$$\leq \mathbb{E}_\xi \sup_{\substack{j \in [q], f \in \mathcal{F} \\ w_1 \ldots, w_q: \|w_j\|_2 = R}} g\left(\sqrt{q \cdot |\sum_{i=1}^m \xi_i \cdot \sigma(w_j^\top f_j(x_i))|^2}\right)$$

$$= \mathbb{E}_\xi \sup_{\substack{j \in [q], f \in \mathcal{F} \\ w: \|w\|_2 = R}} g\left(\sqrt{q} \cdot |\sum_{i=1}^m \xi_i \cdot \sigma(w^\top f_j(x_i))|\right).$$

Since $g(|z|) \leq g(z) + g(-z)$,

$$\mathbb{E}_\xi \sup_{\substack{j \in [q], f \in \mathcal{F} \\ w: \|w\|_2 = R}} g\left(\sqrt{q} \cdot |\sum_{i=1}^m \xi_i \cdot \sigma(w^\top f_j(x_i))|\right)$$

$$\leq \mathbb{E}_\xi \sup_{\substack{j \in [q], f \in \mathcal{F} \\ w: \|w\|_2 = R}} g\left(\sqrt{q} \cdot \sum_{i=1}^m \xi_i \cdot \sigma(w^\top f_j(x_i))\right)$$

$$+ \mathbb{E}_\xi \sup_{\substack{j \in [q], f \in \mathcal{F} \\ w: \|w\|_2 = R}} g\left(-\sqrt{q} \cdot \sum_{i=1}^m \xi_i \cdot \sigma(w^\top f_j(x_i))\right)$$

$$= 2\mathbb{E}_\xi \sup_{\substack{j \in [q], f \in \mathcal{F} \\ w: \|w\|_2 = R}} g\left(\sqrt{q} \cdot \sum_{i=1}^m \xi_i \cdot \sigma(w^\top f_j(x_i))\right),$$

where the last equality follows from the symmetry in the distribution of the $\xi_i$ random variables. By Equation 4.20 in [51] and Cauchy-Schwartz, we have the following:

$$\mathbb{E}_\xi \sup_{\substack{j \in [q], f \in \mathcal{F} \\ w: \|w\|_2 = R}} g\left(\sqrt{q} \cdot \sum_{i=1}^m \xi_i \cdot \sigma(w^\top f_j(x_i))\right)$$

$$\leq \mathbb{E}_\xi \sup_{\substack{j \in [q], f \in \mathcal{F} \\ w: \|w\|_2 = R}} g\left(\sqrt{q} \cdot \sum_{i=1}^m \xi_i \cdot w^\top f_j(x_i)\right)$$

$$\leq \mathbb{E}_\xi \sup_{\substack{j \in [q], f \in \mathcal{F} \\ w: \|w\|_2 = R}} g\left(\sqrt{q} \cdot \|w\|_2 \left\|\sum_{i=1}^m \xi_i \cdot f_j(x_i)\right\|_2\right)$$

$$\leq \mathbb{E}_\xi \sup_{j \in [q], f \in \mathcal{F}} g\left(\sqrt{q} R \left\|\sum_{i=1}^m \xi_i \cdot f_j(x_i)\right\|_2\right).$$

$\square$

**Proposition 3.1.** *Let $G$ be a neural network architecture of depth $L$ and let $\rho > 0$. Let $X = \{x_i\}_{i=1}^m$ be a set of samples. Then,*

$$\mathcal{R}_X(\mathcal{F}_{G,\rho}) \leq \frac{\rho}{m} \cdot \left(1 + \sqrt{2(\log(2)L + \sum_{l=1}^{L-1} \log(\deg(G)_l) + \log(C))}\right)$$

$$\cdot \sqrt{\max_{j_0, \ldots, j_L} \prod_{l=1}^{L-1} |\text{pred}(l, j_l)| \cdot \sum_{i=1}^m \|z_{j_0}^0(x_i)\|_2^2},$$

*where the maximum is taken over $j_0, j_1, \ldots, j_L$, such that, $j_{l-1} \in \text{pred}(l, j_l)$ for all $l \in [L]$.*

*Proof.* We denote by $f_w$ an arbitrary member of $\mathcal{F}_{G,\rho}$ and $w_{j_l}^l$ the weights of the $j_l$th neuron of the $l$th layer. Due to the homogeneity of the ReLU function, each function $f_w \in \mathcal{F}_{G,\rho}$ can be rewritten as $f_{\hat{w}}$, where $\hat{w}^L := \rho \frac{w^L}{\|w^L\|_F}$ and for all $l < L$ and $j_l \in [d_l]$, $\hat{w}_{j_l}^l := \frac{w_{j_l}^l}{\max_{j \in [d_l]} \|w_j^l\|_F}$. In particular, we have $\mathcal{F}_{G,\rho} \subset \hat{\mathcal{F}}_{G,\rho} := \{f_w \mid \|w^L\|_F \le \rho$ and $\forall i < L, j_l \in [d_l]: \|w_{j_l}^l\|_F \le 1\}$ since the ReLU function is homogeneous.

We first consider that

$$
\begin{aligned}
m\mathcal{R} := m\mathcal{R}_X(\hat{\mathcal{F}}_{G,\rho}) &= \mathbb{E}_\xi \left[ \sup_{\hat{w}} \sum_{i=1}^m \sum_{r=1}^C \xi_{ir} f_{\hat{w}}(x_i)_r \right] \\
&= \mathbb{E}_\xi \left[ \sup_{\hat{w}} \sum_{i=1}^m \sum_{r=1}^C \xi_{ir} \sum_{j_{L-1}} \langle \hat{w}_{rj_{L-1}}^L, z_{j_{L-1}}^{L-1}(x_i) \rangle \right] \\
&= \mathbb{E}_\xi \left[ \sup_{\hat{w}} \sum_{j_{L-1}} \sum_{r=1}^C (\hat{w}_{rj_{L-1}}^L)^\top \sum_{i=1}^m \xi_{ir} z_{j_{L-1}}^{L-1}(x_i) \right] \\
&\le \rho \cdot \mathbb{E}_\xi \left[ \sup_{\hat{w}} \max_{r, j_{L-1}} \left\| \sum_{i=1}^m \xi_{ir} z_{j_{L-1}}^{L-1}(x_i) \right\|_2 \right]
\end{aligned}
$$

where the last inequality follows from moving the norm of $\hat{w}^L$ to the vector $\hat{w}_{rj_{L-1}}^L$ for maximizing the term $\sum_{j_{L-1}} \sum_{r=1}^C (\hat{w}_{rj_{L-1}}^L)^\top \sum_{i=1}^m \xi_{ir} z_{j_{L-1}}^{L-1}(x_i)$ and applying the Cauchy-Schwartz inequality. Next, we apply Jensen's inequality,

$$
\begin{aligned}
\mathbb{E}_\xi \left[ \sup_{\hat{w}} \max_{r, j_{L-1}} \left\| \sum_{i=1}^m \xi_{ir} z_{j_{L-1}}^{L-1}(x_i) \right\|_2 \right] &\le \frac{1}{\lambda} \cdot \log \mathbb{E}_\xi \sup_{\hat{w}, r, j_{L-1}} \exp\left( \lambda \left\| \sum_{i=1}^m \xi_{ir} z_{j_{L-1}}^{L-1}(x_i) \right\|_2 \right) \\
&\le \frac{1}{\lambda} \cdot \log \mathbb{E}_\xi \sup_{\hat{w}, r, j_{L-1}} \exp\left( \lambda \sqrt{\left\| \sum_{i=1}^m \xi_{ir} z_{j_{L-1}}^{L-1}(x_i) \right\|_2^2} \right),
\end{aligned}
$$

where the supremum is taken over the weights $\hat{w}_{j_l}^l$ ($l \in [L]$, $j_l \in [d_l]$) that are described above. Next, we use Lemma 3.4,

$$
\begin{aligned}
m\mathcal{R} &\le \frac{\rho}{\lambda} \cdot \log\left( \mathbb{E}_\xi \sup_{\hat{w}, r, j_{L-1}} \exp\left( \lambda \cdot \sqrt{\left\| \sum_{i=1}^m \xi_{ir} \cdot z_{j_{L-1}}^{L-1}(x_i) \right\|_2^2} \right) \right) \\
&= \frac{\rho}{\lambda} \cdot \log\left( \mathbb{E}_\xi \sup_{\hat{w}, r, j_{L-1}} \exp\left( \lambda \cdot \sqrt{\left\| \sum_{i=1}^m \xi_{ir} \cdot \sigma(\hat{w}_{j_{L-1}}^{L-1} v_{j_{L-1}}^{L-2}(x_i)) \right\|_2^2} \right) \right) \\
&\le \frac{\rho}{\lambda} \cdot \log\left( 2\mathbb{E}_\xi \sup_{\hat{w}, r, j_{L-1}} \exp\left( \lambda \cdot \sqrt{\left\| \sum_{i=1}^m \xi_{ir} \cdot v_{j_{L-1}}^{L-2}(x_i) \right\|_2^2} \right) \right) \\
&= \frac{\rho}{\lambda} \cdot \log\left( 2\mathbb{E}_\xi \sup_{\hat{w}, r, j_{L-1}} \exp\left( \lambda \cdot \sqrt{\sum_{j_{L-2} \in \text{pred}(L-1, j_{L-1})} \left\| \sum_{i=1}^m \xi_{ir} \cdot \sigma(\hat{w}_{j_{L-2}}^{L-2} v_{j_{L-2}}^{L-3}(x_i)) \right\|_2^2} \right) \right) \\
&\le \frac{\rho}{\lambda} \cdot \log\left( 4\mathbb{E}_\xi \sup_{\hat{w}, r, j_{L-1}, j_{L-2}} \exp\left( \lambda \cdot \sqrt{|\text{pred}(L-1, j_{L-1})| \left\| \sum_{i=1}^m \xi_{ir} \cdot v_{j_{L-2}}^{L-3}(x_i) \right\|_2^2} \right) \right),
\end{aligned}
$$

where the supremum is taken over the parameters of $f_{\hat{w}}$ and $j_1 \in \text{pred}(L, j_0)$. By applying this process recursively $L$ times, we obtain the following inequality,

$$
m\mathcal{R} \le \frac{\rho}{\lambda} \cdot \log\left( 2^L \mathbb{E}_\xi \max_{r, j_0, \dots, j_L} \exp\left( \lambda \cdot \sqrt{\prod_{l=1}^{L-1} |\text{pred}(l, j_l)| \cdot \left\| \sum_{i=1}^m \xi_{ir} \cdot z_{j_0}^0(x_i) \right\|_2} \right) \right), \tag{5}
$$

where the maximum is taken over $j_0, \ldots, j_L$, such that, $j_{l-1} \in \text{pred}(l, j_l)$ and $r \in [C]$. We notice that

$$\mathbb{E}_\xi \max_{r, j_0, \ldots, j_L} \exp\left(\lambda \cdot \sqrt{\prod_{l=1}^{L-1} |\text{pred}(l, j_l)|} \cdot \left\| \sum_{i=1}^m \xi_{ir} \cdot z_{j_0}^0(x_i) \right\|_2 \right)$$

$$\leq \sum_{r, j_0, \ldots, j_L} \mathbb{E}_\xi \exp\left(\lambda \cdot \sqrt{\prod_{l=1}^{L-1} |\text{pred}(l, j_l)|} \cdot \left\| \sum_{i=1}^m \xi_{ir} \cdot z_{j_0}^0(x_i) \right\|_2 \right) \qquad (6)$$

$$\leq C \prod_{l=1}^{L-1} \deg(G)_l \cdot \max_{r, j_0, \ldots, j_L} \mathbb{E}_\xi \exp\left(\lambda \cdot \sqrt{\prod_{l=1}^{L-1} |\text{pred}(l, j_l)|} \cdot \left\| \sum_{i=1}^m \xi_{ir} \cdot z_{j_0}^0(x_i) \right\|_2 \right).$$

Following the proof of Theorem 1 in [14], by applying Jensen's inequality and Theorem 6.2 in [57] we obtain that for any $\alpha > 0$,

$$\mathbb{E}_\xi \exp\left(\alpha \left\| \sum_{i=1}^m \xi_{ir} \cdot z_{j_0}^0(x_i) \right\|_2 \right) \leq \exp\left(\frac{\alpha^2 \sum_{i=1}^m \|z_{j_0}^0(x_i)\|_2^2}{2} + \alpha \sqrt{\sum_{i=1}^m \|z_{j_0}^0(x_i)\|_2^2} \right). \qquad (7)$$

Hence, by combining equations 5-7 with $\alpha = \lambda \cdot \sqrt{\prod_{l=1}^{L-1} |\text{pred}(l, j_l)|}$, we obtain that

$$m\mathcal{R} \leq \frac{\rho}{\lambda} \cdot \log\left(2^L C \prod_{l=1}^{L-1} \deg(G)_l \cdot \max_{r, j_0, \ldots, j_L} \mathbb{E}_\xi \exp\left(\lambda \cdot \sqrt{\prod_{l=1}^{L-1} |\text{pred}(l, j_l)|} \cdot \left\| \sum_{i=1}^m \xi_{ir} \cdot z_{j_0}^0(x_i) \right\|_2 \right) \right)$$

$$= \frac{\rho}{\lambda} \cdot \max_{j_0, \ldots, j_L} \log\left(2^L C \prod_{l=1}^{L-1} \deg(G)_l \cdot \mathbb{E}_\xi \exp\left(\lambda \cdot \sqrt{\prod_{l=1}^{L-1} |\text{pred}(l, j_l)|} \cdot \left\| \sum_{i=1}^m \xi_{ir} \cdot z_{j_0}^0(x_i) \right\|_2 \right) \right)$$

$$\leq \frac{\rho \cdot (\log(2)L + \sum_{l=1}^{L-1} \log(\deg(G)_l) + \log(C))}{\lambda}$$

$$+ \frac{\lambda \rho \max_{r, j_0, \ldots, j_L} \prod_{l=1}^{L-1} |\text{pred}(l, j_l)| \cdot \sum_{i=1}^m \|z_{j_0}^0(x_i)\|_2^2}{2}$$

$$+ \rho \sqrt{\max_{j_0, \ldots, j_L} \prod_{l=1}^{L-1} |\text{pred}(l, j_l)| \cdot \sum_{i=1}^m \|z_{j_0}^0(x_i)\|_2^2}$$

The choice $\lambda = \sqrt{\frac{2(\log(2)L + \sum_{l=1}^{L-1} \log(\deg(G)_l) + \log(C))}{\max_{j_0, \ldots, j_L} \prod_{l=1}^{L-1} |\text{pred}(l, j_l)| \cdot \sum_{i=1}^m \|z_{j_0}^0(x_i)\|_2^2}}$ gives the desired inequality. $\qquad \square$

**Theorem 3.2.** *Let $P$ be a distribution over $\mathbb{R}^{c_0 d_0} \times \{\pm 1\}$. Let $S = \{(x_i, y_i)\}_{i=1}^m$ be a dataset of i.i.d. samples selected from $P$. Then, with probability at least $1 - \delta$ over the selection of $S$, for any $f_w \in \mathcal{F}_G$,*

$$\text{err}_P(f_w) - \text{err}_S^\gamma(f_w) \leq \frac{2\sqrt{2}(\rho(w) + 1)}{\gamma m} \cdot \left(1 + \sqrt{2(\log(2)L + \sum_{l=1}^{L-1} \log(\deg(G)_l) + \log(C))} \right)$$

$$\cdot \sqrt{\max_{j_0, \ldots, j_L} \prod_{l=1}^{L-1} |\text{pred}(l, j_l)| \cdot \sum_{i=1}^m \|z_{j_0}^0(x_i)\|_2^2} + 3\sqrt{\frac{\log(2(\rho(w) + 2)^2/\delta)}{2m}},$$

*where the maximum is taken over $j_0, \ldots, j_L$, such that, $j_{l-1} \in \text{pred}(l, j_l)$ for all $l \in [L]$.*

*Proof.* Let $t \in \mathbb{N} \cup \{0\}$ and $\mathcal{G}_t = \mathcal{F}_{G,t}$. By Lemma 2.2, with probability at least $1 - \frac{\delta}{t(t+1)}$, for any function $f_w \in \mathcal{G}_t$, we have

$$\text{err}_P(f_w) - \text{err}_S^\gamma(f_w) \leq \frac{2\sqrt{2}}{\gamma} \cdot \mathcal{R}_X(\mathcal{G}_t) + 3\sqrt{\frac{\log(2/\delta)}{2m}}. \qquad (8)$$

By Proposition 3.1 in the main text, we have

$$\mathcal{R}_X(\mathcal{G}_t) \leq \frac{t}{m} \cdot \left(1 + \sqrt{2(\log(2)L + \sum_{l=1}^{L-1} \log(\deg(G)_l) + \log(C))}\right)$$
$$\cdot \sqrt{\max_{j_0,\ldots,j_L} \prod_{l=1}^{L-1} |\mathrm{pred}(l, j_l)| \cdot \sum_{i=1}^{m} \|z_{j_0}^0(x_i)\|_2^2},$$

because of the union bound over all $t \in \mathbb{N}$, equation 1 holds uniformly for all $t \in \mathbb{N}$ and $f_w \in \mathcal{G}_t$ with probability at least $1-\delta$. For each $f_w$ with norm $\rho(w)$ we then apply the bound with $t = \lceil \rho(w) \rceil$ since $f_w \in \mathcal{G}_t$, we have

$$\mathrm{err}_P(f_w) - \mathrm{err}_S^\gamma(f_w) \leq \frac{2\sqrt{2} \cdot t}{\gamma m} \cdot \left(1 + \sqrt{2(\log(2)L + \sum_{l=1}^{L-1} \log(\deg(G)_l) + \log(C))}\right)$$
$$\cdot \sqrt{\max_{j_0,\ldots,j_L} \prod_{l=1}^{L-1} |\mathrm{pred}(l, j_l)| \cdot \sum_{i=1}^{m} \|z_{j_0}^0(x_i)\|_2^2} + 3\sqrt{\frac{\log(2(t+1)^2/\delta)}{2m}}$$
$$\leq \frac{2\sqrt{2}(\rho(w) + 1)}{\gamma m} \left(1 + \sqrt{2(\log(2)L + \sum_{l=1}^{L-1} \log(\deg(G)_l) + \log(C))}\right)$$
$$\cdot \sqrt{\max_{j_0,\ldots,j_L} \prod_{l=1}^{L-1} |\mathrm{pred}(l, j_l)| \sum_{i=1}^{m} \|z_{j_0}^0(x_i)\|_2^2} + 3\sqrt{\frac{\log(2(\rho(w) + 2)^2/\delta)}{2m}},$$

which proves the desired bound. $\square$

