# OpenReview forum: "Norm-based Generalization Bounds for Sparse Neural Networks"
_NeurIPS.cc/2023/Conference — NeurIPS 2023 poster_

### Official Review · Reviewer_G4im · 2023-07-05

**Soundness:** 3 good
**Presentation:** 3 good
**Contribution:** 3 good
**Rating:** 5
**Confidence:** 2

**Summary:**

This paper presents the derivation of generalization bounds for sparsely connected neural networks. The authors extend the peeling arguments to establish a bound on the Rademacher complexity of these networks based on the number of connections between each neuron and its predecessors and norms of the network weights. The proposed bound is then applied to the specific case of Convolutional Neural Networks (CNNs), and the authors validate their bounds through experimental evaluations.

**Strengths:**

1) The paper demonstrates a clear and well-structured writing style.
2) The related work is thoroughly cited, indicating a comprehensive understanding of the existing literature.
3) The extended approach presented in the paper is interesting.

**Weaknesses:**

1) The provided bounds exhibit exponential scaling in the network depth, even when disregarding the exponential dependence of $\rho(w)$ on depth.

2) The nature of the bound shares similarities with parameter counting bounds, as it explicitly relies on the number of connections between each neuron and its predecessors (e.g., filter size in CNNs). This explicit dependence remains present even if the learned network deviates minimally from its initialization.

3) In comparisons of related work, the authors use unrealistic cases to demonstrate the superiority of their approach. For instance,  input dimension is 2^L, filter size =2.

**Questions:**

1) Is it possible to adapt the proof technique to mitigate or eliminate the exponential dependence on $\rho(w)$ (e.g.,  Lipschitz augmentation techniques mentioned in [1])?

2) Can the explicit dependence on the filter size (number of connections) be replaced with an implicit dependence, for example, through incorporating weight norms?

[1] Wei, C., & Ma, T. (2019). Data-dependent sample complexity of deep neural networks via Lipschitz augmentation. Advances in Neural Information Processing Systems, 32.


**Limitations:**

Yes

---

> ### Author Rebuttal · Authors · 2023-08-08
>
> > Reviewer: The provided bounds exhibit exponential scaling in the network depth, even when disregarding the exponential dependence of $\rho(w)$ on depth.
>
> > The nature of the bound shares similarities with parameter counting bounds, as it explicitly relies on the number of connections between each neuron and its predecessors (e.g., filter size in CNNs). This explicit dependence remains present even if the learned network deviates minimally from its initialization.
>
> Answer: While it may initially seem that our bound displays a dependence on parameter counting, we want to highlight that it is not always the case. In traditional norm-based bounds (e.g., Golowich et al., Bartlett et al.) the bounds typically scale with $(C(w) \cdot \max\_i||x\_{i}||)/\sqrt{m}$, where $C(w)$ represents a measure of complexity (e.g., the product of the norms of the weight matrices). In our setting, we do introduce an additional term $\prod^{L}\_{l=1} k\_{l}$, but rather than taking the maximal norm over the samples, $\max\_{i}||x\_{i}||$, our bound scales with the maximal norm over patches, $\max\_{i,j}||z\_{j}(x\_{i})||$.
>
> As we discuss in lines 224-237, if the patches within the various samples are relatively balanced and non-overlapping, then $\sqrt{\prod^{L}\_{l=1} k\_{l} \cdot \max\_{i,j}||z\_{j}(x\_{i})||^2}$ is proportional to $\max\_{i}||x\_i||$. Consequently, what might initially appear as an exponential dependence on depth via $\prod^{L}\_{l=1} k\_{l}$ is not accurate because $\max\_{i,j}||z\_{j}(x_i)||^2$ is much smaller than $\max\_{i}||x\_{i}||^2$. Then, the only thing left to differ between our bound and Golowich’s bound is the notion of complexity $C(w)$. In our bound it measures the product of the norms of the parameter kernels at the various layers and in Golowich’s bound it is the product of the norms of the linear transformations associated with the convolutional layers. As we discuss in lines 62-70, this is significantly larger than our measure of complexity.
>
> > Reviewer: In comparisons of related work, the authors use unrealistic cases to demonstrate the superiority of their approach. For instance, input dimension is $2^L$, filter size =2.
>
> Answer: The choice of $d_\{0}=2^L$ is purely technical to illustrate the scales of the bound with respect to depth $L$. However, the same logic holds for an arbitrary choice of $d_\{0}$ and any kernel size $k$. For example, we can still take a network with $L_0=\lfloor log\_\{k}(d_0) \rfloor$ convolutional layers with kernel sizes $k$ and multiple fully connected layers on top of them. Then, $\prod^{L_0}\_\{l=1} k\_\{l} =k^{L_0}$, each term $z_\{j}$ will be a patch of two pixels. There are $\lfloor d_\{0}/k \rfloor = k^{L_0}$ such patches. Therefore, we have $\max\_\{j}\sum\_\{i}||z\_\{j}(x_i)||^2 \cdot  k^{L_0} \leq \beta \cdot \sum\_\{i}||x\_i||^2$ assuming the patches are $\beta$-balanced.
>
> > Reviewer: Is it possible to adapt the proof technique to mitigate or eliminate the exponential dependence on $\rho(w)$ (e.g., Lipschitz augmentation techniques mentioned in [1])?
>
> Answer: That would be a very interesting future direction. Generally speaking, relaxing the dependence on Frobenius norms would be useful for making our bounds more practical. However, at this point we are not sure if it would be possible because the results in [1] rely on analyzing the covering number of the function class in order to derive a bound for the Rademacher complexity. Our analysis, on the other hand, relies on a repeated application of the peeling lemma without using covering numbers.
>
> Regarding the scaling of $\rho(w)$, we agree that it could potentially exponentially grow with $L$. However, it is not necessarily the case. Suppose we have $L-1$ layers with norms $(1+1/L)^{2}$ and one layer of norm $5(1+1/L)^{2}$, then $\rho \approx 5e^2$.
>
> > Reviewer: Can the explicit dependence on the filter size (number of connections) be replaced with an implicit dependence, for example, through incorporating weight norms?
>
> Answer: It would be a very interesting research direction to explore to what extent we can replace the filter sizes with an implicit dependence. We think that one could potentially relax the dependence on the size of the predecessor set by taking into account the norm of the connections with the neurons in the predecessor sets. At least it might be possible to avoid counting neurons in predecessor sets if their weights are very small. However, it is still something we are exploring and has yet to be formalized.
>
> We would like to reiterate that despite the existence of the term $\sqrt{\prod^{L}\_{l=1} k\_{l}}$, our bound also scales with $\max_{i,j}||z_{j}(x_i)||$ instead of $\max\_{i}||x\_{i}||$. As we discussed above, in some cases (see lines 224-237), the expression $\max\_{i}||x\_{i}||$ scales as $\sqrt{\prod^{L}\_{l=1} k\_{l}}$ times the expression $\max_{i,j}||z_{j}(x_i)||$. In such cases, $\sqrt{\prod^{L}\_{l=1} k\_{l} \cdot \max\_{i,j}||z\_{j}(x_i)||^2}$ is proportional to $\max\_{i}||x\_{i}||$ and the bound scales as $\rho(w) \cdot \max\_{i} ||x\_\{i}||/\sqrt{m}$ which is independent of the filter sizes.

---

> > ### Comment · Reviewer_G4im · 2023-08-20
> >
> > Thanks for the thoughtful rebuttal. However, my concerns about the exponential depth were not addressed adequately. The main argument of the authors assumes that the number of image patches is exponential in the number of convolutional layers; this is very limiting (consider Resnet-18/50 with filter size $3\times 3$).
> >
> > I believe extending the peeling argument to CNNs or general sparsely connected networks can be of theoretical interest. Nevertheless, currently, other approaches in the literature seem to provide better bounds in more realistic settings (e.g., ones based on covering numbers). I will therefore maintain my borderline score.

---

### Official Review · Reviewer_dgpJ · 2023-07-05

**Soundness:** 3 good
**Presentation:** 4 excellent
**Contribution:** 3 good
**Rating:** 7
**Confidence:** 3

**Summary:**

The authors present a bound on the Rademacher complexity of sparse neural networks, which include in particular convolutional neural networks. This bound improves over previously-known bounds for CNNs, and applies more broadly since it does not assume weight sharing, but only sparsity. The bounds are illustrated by numerical experiments.

**Strengths:**

The paper tackles the important question of the understanding of the advantages of the convolution structure from a statistical point of view. The proposed bounds improve over known bounds by leveraging modern proof techniques for the Rademacher complexity of deep neural networks. The paper is well-written and easy to follow. I checked the proofs partially, and the mathematical details are globally well-written as well.

**Weaknesses:**

I am not convinced by the numerical experiments as they are, because I do not think they back the claim of the authors on the tightness of their generalization bound. There are several orders of magnitude between the bound and the reported generalization gap. Of course, I am well aware of the difficulty of decreasing this spread, but I think it would be interesting to complement the experiments to prove the point by
+ reporting the value of other generalization bounds in the same setting, in order to illustrate that the proposed bound is indeed tighter than others.
+ varying the amount of regularization (I guess, the lambda parameter) described in lines 334-342. If rho(w) does indeed play a role in the generalization power of the neural network, I think you should observe a negative correlation between the regularization strength and the generalization gap. This would be more interesting than just reporting the value of rho(w) as in Figure 1, since this value in itself does not have a strong meaning.

**Questions:**

+ Line 124 (and elsewhere in the paper): you mention that you study the overparametrized regime, but your bounds hold independently of the under/overparameterization. I suggest to rephrase to avoid confusion.
+ Lines 126-129: the interpretation of Rademacher complexity is misleading. It is not the expected performance of the class, but of the worse case error in the class. This is very different.
+ Line 161: shouldn’t it be v_i^l and pred(l, i) instead of v_i^{l-1} and pred(l-1,i)? To fit with the definition of v_i^l two lines above.
+ Line 197 and 205: I find the numbering of j_1 … j_L extremely confusing, I think it would be much more intuitive to reverse the order, in order to make pred(l, j_l) appear in the formula instead of pred(l, j_{L-l}). There is a related issue in the proof, line 134 of the Appendix, I believe it should be j_{L-l+1} instead of j_{l+1}. Also, an explanation on what’s going on (and a drawing?) would be most welcome. You’re basically enumerating the paths in the graph from the input to the output, right?
+ Lines 261 to 282: ok, but I believe these explanations are not specific to your bound, but are true for any norm-based bound. This should be written more explicitely.
+ Line 274: I don’t think the bound of Theorem 3.2 is equal to 0 if rho(w)=0? Could you elaborate on this?


**Limitations:**

See weaknesses.

---

> ### Author Rebuttal · Authors · 2023-08-08
>
> > Reviewer: reporting the value of other generalization bounds in the same setting, in order to illustrate that the proposed bound is indeed tighter than others.
>
> Answer: Following the reviews we conducted multiple experiments to compare our bounds with other generalization bounds for convnets that were suggested by the reviewers. We consistently observe that our bound is much tighter than the other bounds. Please refer to the comments to all of the reviewers for details and the pdf for the plots.
>
> > Reviewer: varying the amount of regularization (I guess, the lambda parameter) described in lines 334-342. If rho(w) does indeed play a role in the generalization power of the neural network, I think you should observe a negative correlation between the regularization strength and the generalization gap. This would be more interesting than just reporting the value of rho(w) as in Figure 1, since this value in itself does not have a strong meaning.
>
> Answer: Following the reviews, we conducted multiple experiments to validate this correlation. We observe that when increasing $\lambda$, our bound decreases (since $\rho$ decreases) as well as the generalization gap.
>
> > Reviewer: Line 124 (and elsewhere in the paper): you mention that you study the overparametrized regime, but your bounds hold independently of the under/overparameterization. I suggest to rephrase to avoid confusion.
>
> Answer: We agree and will rephrase it accordingly.
>
> > Reviewer: Lines 126-129: the interpretation of Rademacher complexity is misleading. It is not the expected performance of the class, but of the worse case error in the class. This is very different.
>
> Answer: We agree and we willl correct this issue in the next version of the paper.
>
> > Reviewer: Line 161: shouldn’t it be $v^l\_\{i}$ and $pred(l, i)$ instead of $v^{l-1}\_\{i}$ and $pred(l-1,i)$? To fit with the definition of $v^l\_\{i}$ two lines above.
>
> Answer: Yes, we will fix it in the next version of the paper.
>
> > Reviewer: Line 197 and 205: I find the numbering of $j\_1 … j\_L$ extremely confusing, I think it would be much more intuitive to reverse the order, in order to make $pred(l, j\_l)$ appear in the formula instead of $pred(l, j\_{L-l})$. There is a related issue in the proof, line 134 of the Appendix, I believe it should be $j\_{L-l+1}$ instead of $j\_{l+1}$. Also, an explanation of what’s going on (and a drawing?) would be most welcome. You’re basically enumerating the paths in the graph from the input to the output, right?
>
> Answer: We will reverse the order for simplicity and yes we are essentially enumerating the paths.
>
> > Reviewer: Lines 261 to 282: ok, but I believe these explanations are not specific to your bound, but are true for any norm-based bound. This should be written more explicitly.
>
> Answer: We will make it more explicit in the next version of the paper.
>
> > Reviewer: Line 274: I don’t think the bound of Theorem 3.2 is equal to 0 if rho(w)=0? Could you elaborate on this?
>
> Answer: That’s true, the bound in Theorem 3.2 would not be zero. It is a honest mistake since the bound in Corollary 3.3. We will fix this issue in the next version of the paper.

---

> > ### Comment · Reviewer_dgpJ · 2023-08-11
> >
> > I thank the authors for taking the time to write the rebuttal. All my questions are addressed thoroughly, in particular thanks to the added experiments that are very convincing. I have updated my rating accordingly.

---

> > > ### Author Response · Authors · 2023-08-13
> > >
> > > Thank you very much for your further appreciation of the paper!

---

### Official Review · Reviewer_LtGn · 2023-07-06

**Soundness:** 4 excellent
**Presentation:** 3 good
**Contribution:** 3 good
**Rating:** 7
**Confidence:** 4

**Summary:**

This work proposes an adaptation of Radamacher complexity analysis for deep neural networks specifically suited to those that are compositionally sparse, as in convolutional architectures. The authors characterize the specific model parametrization by a directed acyclic graph, and provide an upper bound to the Radamacher complexity of neurons that have only a few parents. By adapting the required lemmas to these cases, the authors provide a bound on the generalization gap that scales better with the norm of the weights for convolutional networks. Lastly, the authors demonstrate their bounds in practice for a couple of cases (CIFAR and MNIST).

**Strengths:**

- The majority of the generalization bounds for neural network does not make explicit use of the structure in convolutional filters, and this paper seeks to fill this important gap
- The numerical evaluation of their bounds is highly appreciated
- The writing and presentation are clear

**Weaknesses:**

- To my understanding, some claims are overstated or incorrect (see below).
- Some further numerical evaluation would support their claims better (see below).

**Questions:**

I liked this paper, and enjoyed reading it very much.

1. My main concerns regard the authors depiction of the scaling of their bound with the depth of the network, L: when discussing the implications of theorem 3.2 (main result), the authors conclude that their bound is $\mathcal O( \rho \sqrt{ L \beta / m})$, and they therefore comment that their bound has a mild dependency on depth $O(\sqrt{L})$. While I might be missing something, to my understanding this is incorrect for two independent reasons:

    a) the bound on the network norm, $\rho$, scales exponentially with $L$ in general (line 185). It is true that we don’t know what the correct scaling for this norm is, but since this norm is the product of Frobenius norms (of the filters), the only way for $\rho$ not to grow exponentially with $L$ is for these norms to decrease exponentially with $L$. Naturally, this is not a reasonable assumption (unless the authors can demonstrate this numerically). Their bound, like all other works based on Radamacher complexities, have an exponential dependence on L.

    b) on line 212, the authors explain that for some "constant" $\beta$, one can upper bound $\prod^L k_l max_j \sum_m ||z_j(x_i)||^2 \leq \beta \sum_m ||x_i||^2$. I do not believe it is possible for this upper bound to hold for a constant beta (i.e., $\beta$ that is independent of $L$), right? If $L$ increases, then necessarily $\beta$ should increase (and exponentially so) for the bound to hold.

    Therefore, by presenting their overall bound as $O( \rho \sqrt{ L \beta / m})$, the dependence on L seems incorrect. I’d be happy to learn what I am missing if I'm incorrect about these points above.


1. The inclusion of numerical results that evaluate the obtained bounds is highly appreciated. While it is true that these bounds seem interesting and beginning to be reasonable, they do not provide non-vacuous bounds as stated in the abstract. I believe this should be rephrased. Indeed, in the conclusions the authors do mention, accurately, that their bounds are not yet practical.

1. I am curious about how the authors choose the value for the network norm, $\rho(w)$, in order to evaluate their bound. As presented, in Theorem 3.2, this value characterizes the hypothesis class and is fixed (i.e. chosen independent of data). In the experiments, the authors mention that they control the value of this norm via regularization, and obtain trained model $w*$. But then, what value for $\rho(w)$ is chosen at the end to instantiate the bound? Naturally, the value $\rho(w*)$ cannot be chosen, as it is data dependent. Alternatively, the authors could extend their results to hold uniformly over the choice of $\rho$, but this comes at the expense of an additional statistical cost (which I don’t believe the authors are taking into account). Could you clarify?

 1. The authors have done a reasonable job of comparing with existing literature in generalization theory. However, the sensitivity analysis/(modified layer-peeling process) bares resemblance to prior work (such as Singla et al) that studies the Lipschitz constant of convolutional networks. I encourage the authors to cite these works as well.

2. The authors contend that the computed bounds are non-uniform as they rely on the norm-based measure $\rho(w)$. However, the bounds *are* uniform for all similarly norm-bounded networks. This presents a limitation of the Rademacher analysis as the worst-case deviation between networks with $\rho(w) < \rho*$ could be much larger than the generalization gap of the trained predictor. While this phenomena is not specific to this work, I would encourage the authors to include/contextualize this in their discussion of vacuous bounds.

**Smaller observations and typos:**

i) In Eq.(1), the (-0.5) exponent should be (0.5).

ii) The numerical results (Figs 1 and 2) seem out of place on Page 5. Please move closer to Sec. 4.

iii) \mathcal{F}_G should be \mathcal{F}_{G,\rho} in line 204

iv) It’s unclear to me why the authors stress so much that their bounds also hold when the network perfectly fits the data. This is true, but not particularly enlightening given that the networks they use, in practice, do not.

v) “we rewrite right-hand” → “we rewrite the right-hand..”

vi) The sentence “Furthermore, since our bound…” seems incomplete.

vii) In the last sentence of their paper, the authors state that “..our experiments show that they are quite tight for simple classification problems, unlike other bounds based on parameter counting, suggesting that the underlying theory is sound and does not need a basic reformulation.” I have troubles with this, since every paper that states “this theory will never need a reformulation” is wrong. It’s clear that this paper makes some nice contribution to understand generalization in deep networks, but the claim seems an exaggeration (e.g., it suffices to note that the bounds are still vacuous!).

viii) Subjective: The notation z_j^l(x) is a bit unnatural to me, as z_j^l was defined as node in a DAG instead of a function of the input. Maybe a footnote would clarify this.

ix) in line 127, the authors colloquially defined the Radamacher complexity of a function class as “the expected performance of the class on random data”. I understand that this is just a colloquial description, but it’s not quite accurate: this notion of complexity measures the *maximal* correlation of any function in the function class to random labeling of the empirical sample.

x) line 145: “om” → “on”?

xi) line 134: “the Rademacher complexity has the added advantage that it is data-dependent and can be measured from finite samples.” This is incorrect (unless the authors can solve the maximization over all f in the function class). What the authors likely meant is that it can be upper bounded based on a finite sample.

**Limitations:**

Their limitations have received little attention. It would be beneficial to expand on these. For example, their bounds (as they do mention) are not yet practical. What do the authors recognize as the main limitation in their analysis that prevents this from hapening?

---

> ### Author Rebuttal · Authors · 2023-08-08
>
> > Reviewer: “the bound on the network norm, $\rho$, scales exponentially with $L$ in general (line 185)... since this norm is the product of Frobenius norms (of the filters), the only way for $\rho$ not to grow exponentially with $L$ is for these norms to decrease exponentially with $L$...”
>
> Answer: We agree that $\rho$ could potentially grow exponentially with $L$. The main contribution of the paper is not to improve the dependence of generalization bounds on depth, rather to better understand the role of sparsity in generalization. Just to note, $\rho$ does not necessarily either grows exponentially with $L$ or the weight norms decay exponentially with $L$. For example, $\rho \approx e$ when each layer  is norm $(1+1/L)$.
>
> > Reviewer: “on line 212… "constant" $\beta$, one can upper bound $\prod^L k_l \max_\{j} \sum_\{i} ||z_\{j}(x_\{i})||^2 \leq \beta \sum_\{i} ||x_i||^2$. I do not believe it is possible for this upper bound to hold for a constant $\beta$”
>
> Answer: The constant $\beta$ is designed to be independent of $L$. Specifically, it evaluates the highest ratio between the norm of the most intense pixel and the average pixel norm within an image. As this measure solely relies on the training samples and is not associated with the network architecture, it doesn't depend on $L$. We'll make sure to clarify this point in the next version of our paper.
>
> > Reviewer: While it is true that these bounds seem interesting and beginning to be reasonable, they do not provide non-vacuous bounds as stated in the abstract. I believe this should be rephrased. Indeed, in the conclusions the authors do mention, accurately, that their bounds are not yet practical.
>
> Answer: Indeed, we do not argue that the bounds are practical at this point and agree that they are beginning to be reasonable. We will rephrase these statements to make it clearer that the bounds are not yet practical.
>
> > Reviewer: “I am curious about how the authors choose the value for the network norm, $\rho(w)$, in order to evaluate their bound… but this comes at the expense of an additional statistical cost (which I don’t believe the authors are taking into account)...”
>
> Answer: The norm $\rho(w)$ is actually not fixed in Theorem 3.2. For each function $f_w$ we have a different norm $\rho(w)$ without selecting $\rho$ before seeing the data. Therefore, by selecting $w$ dependent on the data, $\rho(w)$ could very much also be dependent on the data. To do so, we used a simple trick that is often repeated in the literature (see the proof of Lemma A.9 in Bartlett et al. 2017, pages 22-23). This comes with a minor cost (the bound scales with $\rho+1$ instead of $\rho$).
>
> The idea is as follows: for each positive integer $t$, we combine Lemma 2.2 and Proposition 3.1 for $\rho=t$, $\mathcal{F}\_{G,\rho}=\mathcal{F}\_{G,t}$ and $\delta\_t = \delta/(t(t+1))$ (see Eqs. 6-7 in the Appendix). This gives us a bound that holds for all functions of norm at most $t$. By taking union bound across all $t \in \mathbb{N}$, these bounds hold simultanously with prob $\geq 1-\delta = 1-\sum_{t}\delta/(t(t+1))$. Then, (with probability at least $1-\delta$) for each $f_w$ any bound with $t \geq \rho(w)$ should hold. We choose the bound with $t=\lceil \rho(w) \rceil \leq \rho(w)+1$. Then, with probability at least $1-\delta$, for each $f_w$, the inequality in Theorem 3.2 holds.
>
> > Reviewer: “... the sensitivity analysis/(modified layer-peeling process) bares resemblance to prior work (such as Singla et al)... I encourage the authors to cite these works as well.”
>
> Answer: We are happy to cite and discuss these papers.
>
> > Reviewer: “The authors contend that the computed bounds are non-uniform as they rely on the norm-based measure $\rho(w)$. However, the bounds are uniform for all similarly norm-bounded networks…”
>
> Answer: Please refer to the comment above about adjusting the bounds to depend on $\rho(w)$ instead of a pre-selected $\rho^*$.
>
> > Reviewer: Multiple typos.
>
> Answer: We will incorporate these suggestions in the next version of the paper.
>
> > Reviewer: It’s unclear to me why the authors stress so much that their bounds also hold when the network perfectly fits the data. This is true, but not particularly enlightening given that the networks they use, in practice, do not.
>
> Answer: We will remove these statements.
>
> > Reviewer: “In the last sentence of their paper, the authors state that… does not need a basic reformulation...”
>
> Answer: We agree and will remove this statement.
>
> > Reviewer: “in line 127, the authors colloquially defined the Radamacher complexity of a function class as “the expected performance of the class on random data”... not quite accurate…”
>
> > line 134: “the Rademacher complexity has the added advantage that it is data-dependent and can be measured from finite samples.” This is incorrect…
>
> Answer: We agree and will modify these statements accordingly.
>
> > Reviewer: “Their limitations… For example, their bounds… not yet practical…. main limitation in their analysis…”
>
> Answer: We will expand on the limitations section. The goal of this work in not to provide a quantitatively good estimator of generalization. After all, there are much better ways to estimate generalization. The main goal is to better understand the role of sparsity in generalization, the success of convolutional networks and whether weight sharing is necessary or not.
>
> While improving the bounds could be a challenging, we think that these bounds could be improved by thinking of ways to replace the Frobenius norms with spectral norms or smaller types of norms. Another approach could be deriving bounds that are data-dependent. Most of the bounds in the literature measure the “worst case” norm of the learned function quantified by the product between the complexity of the function (e.g., $\rho(w)$) and the largest norm of a training sample (e.g., $\sup ||x||$). This could be potentially relaxed by quantifying the complexity of the network on the specific training data.

---

> > ### Comment · Reviewer_LtGn · 2023-08-14
> > **Thanks for your answers, and follow up**
> >
> > Dear authors,
> >
> > Thank you for considering my comments, which have clarified most of my questions. I have some follow ups:
> >
> > 1. I respectfully disagree: while I fully understand that the goal of this paper is to understand the role of sparsity resulting from the convolutional (or other-wise sparse) nature of CNNs, the dependence on depth as stated in the paper is incorrect. Indeed, the authors even refer to this on line 69, saying that their bound scales as $\mathcal O(\prod_{i=1}^L ||w_i||_F)$ However, on line 259 they say that their bound scales as $\mathcal O(\sqrt{L})$. So, unless you limit the hypothesis class to weights with norms that decrease exponentially with $L$, this does not make sense to me. Furthermore, I also do not think that $\beta$ can be regarded as a constant independent of $L$. I understand that this is a fixed quantity that can be determined once you fix the architecture, but this quantity must grow with $L$, which is a hyper-parameter of the function class.
> >
> > I appreciate your responses to my other questions - they are clear. As a final note, there's a recent and seemingly related work that the authors might want to look at [Muthukumar et al, Sparsity-aware generalization theory for deep neural networks, COLT 2023] and comment on, given the potential similarity. Of course, as far as I'm concerned, that paper is concurrent to this one, so it has no effect on the evaluation of the current manuscript.

---

> > > ### Author Response · Authors · 2023-08-15
> > > **Thank you for the continued engagement with our work**
> > >
> > > We would like to thank the reviewer for their support and continued engagement with our work!
> > >
> > > > Reviewer: while I fully understand that the goal of this paper is to understand the role of sparsity resulting from the convolutional (or other-wise sparse) nature of CNNs, the dependence on depth as stated in the paper is incorrect.
> > >
> > > Answer: Thanks for the clarification, this is a good point. We will make it clearer in the next version of the paper that besides the term $\sqrt{L}$ there are additional terms in the bound depend on $L$ (e.g., $\rho(w)$) and discuss their relationship with $L$.
> > >
> > > > Reviewer: I also do not think that $\beta$ can be regarded as a constant independent of $L$.
> > >
> > > Answer: Thanks for the concern. We would like to clarify that $\beta$ is defined in line 211 as a number that satisfies $\max\_{j} ||x_{ij}||^2 \leq \beta \cdot Avg\_{j} ||x_{ij}||^2$ for all $i \in [m]$. In particular, $\beta$ is essentially defined as a variable that satisfies $\beta \geq \max\_{i} [(\max\_{j} ||x_{ij}||^2) / (Avg\_{j} ||x_{ij}||^2)]$ which is dependent only on the data (as can be seen from this inequality).
> > >
> > > We note that the definition of $\beta$ implies the inequality in line 212 for the specific setting and architecture described in lines 208-209. Importantly, while the inequality in line 212 holds for the setting outlined in lines 208-209, it should not be true in general. We completely agree with the reviewer that if we would define $\beta$ according to the inequality in line 212 then $\beta$ would have to be dependent on $L$ by definition. But since we define $\beta$ according to line 211, $\beta$ is independent of $L$ and the inequality in line 212 is limited to the setting of 208-209. We will make these points clearer in the next version of the paper.
> > >
> > > To see why the equation in line 212 holds for the setting in lines 208-209:
> > >
> > >
> > > $\prod^{L}\_{l=1} k_l \cdot \max\_{j} \sum^{m}\_{i=1} ||z^0\_{j}(x_i)||^2$
> > >
> > > $= d_0 \max\_{j} \sum^{m}\_{i=1} ||x\_{ij}||^2 $ (follows from $\prod^{L}\_{l=1} k_l = 2^L = d_0$ in lines 208-209)
> > >
> > > $\leq \sum^{m}\_{i=1} d\_{0} \max\_{j} ||x\_{ij}||^2$ (since max sum $\leq sum max)
> > >
> > > $\leq \sum^{m}\_{i=1} d\_{0} \beta Avg\_{j} ||x\_{ij}||^2$ (by the definition of $\beta$ in line 211)
> > >
> > > $= \sum^{m}\_{i=1} d_0 \beta \cdot (1/d_0) \sum\_{j} ||x\_{ij}||^2$
> > >
> > > $= \beta  \sum^{m}\_{i=1} \sum\_{j} ||x_{ij}||^2$
> > >
> > > $= \beta  \sum^{m}\_{i=1} ||x_{i}||^2$ (since $||x_i||^2 = \sum_j ||x_{ij}||^2$)
> > >
> > > > Reviewer: As a final note, there's a recent and seemingly related work that the authors might want to look at... and comment on, given the potential similarity.
> > >
> > > Answer: Thank you for bringing this very interesting paper to our attention. We are happy to add a reference with a discussion of this paper in the next version. We have the impression that the mentioned paper has some similarities with our paper but is still very different in contribution and proof techniques.
> > >
> > > When estimating their bound for the convnets we used, the KL divergence in Theorem 11 seems to be highly correlated with parameter counting. Consider a convnet with $L$ layers. We denote by $c_l$ the number of output channels, $e_l$ the kernel size, and $h_l$ is the height/width of the image output of the $l$th conv layer. We denote by $w_l$ the weight tensor of the layer and by $W_l$ the weight matrix of the linear transformation associated with the conv layer.
> > >
> > > When applying Theorem 11 with the Gaussian prior described in their paragraph “Dependence On Depth” with $\sigma_{sparse}$ and the standard $h_{prior}=0$ which is a very standard approach. We get that the KL divergence term scales as $\sum^{L}\_{l=1} ((d_{eff}+\log(L)) ||W_l||^2_F)/\eta^2_l$.
> > >
> > > Let us analyze $Q = \sum^{L}\_{l=1} (d_{eff} \cdot ||W_l||^2_F)/(\eta^2_l)$. If we choose $s_l=0$ and $\eta_l=1$ we obtain, $d_{eff} = \max_l d_l \log(d_l)$ (see their footnote 10). We note that $d_l=c_{l} \cdot h^2_l$. This means that the KL divergence is at least $\max_l d_l \cdot \sum^{L}\_{l=1} ||W_l||^2_F$. We note that our bound depends on $||w_l||_F$ (the norms of the weight tensor) and not on $||W_l||^2_F = ||w_l||^2_F \cdot h^2_l$. This essentially gives $Q$ that scales as $\max\_{l} (c_l h^2_l) \cdot \sum^{L}\_{l=1} (||w_l||^2_F \cdot h^2_l)$.
> > >
> > > They also mention that $d_{eff}$ can be made smaller than $\max_l d_l$ with different choices of $s_l$. When taking into account the sparsity of the matrix $W_l$ we can choose $s_l$ to be $e^2_l \cdot c_{l-1}$ and $\eta_l$ to be a bound on $||w_l||_F$. Hence, $||W\_{l}||^2_F/\eta^2\_{l} =O (h^2_l)$. However, even then $d\_{l} - s\_{l} = h^2\_{l} c\_{l} - e^2\_{l} c\_{l-1}$ and $d\_{eff}$ is approximately $O(\max\_{l} h^2\_{l} c_l)$ (when $c\_{l}=c\_{l-1}$ and $e\_{l} << h\_{l}$). This gives that $Q$ scales as $(\max\_{l} c\_{l} h^2\_{l}) \sum^{L}\_{l=1} h^2\_{l}$.
> > >
> > > For example, in a CONV-4-500, $c_1=1$, $c_1=c_2=c_3=500$, $e_l=2$, $h_l=\lfloor 28/2^l \rfloor$ and $(\max\_l c_l h^2_l) \sum^{L}\_{l=1} h^2_l \approx 2m$.

---

> > > > ### Comment · Reviewer_LtGn · 2023-08-19
> > > > **Thank you for the added comments**
> > > >
> > > > I thank the authors for their further clarifications. Including these extra comments and clarifications (w.r.t dependence on rho, related works, etc) will increase the quality of this work.

---

> > > > > ### Author Response · Authors · 2023-08-19
> > > > > **Thank you!**
> > > > >
> > > > > We would like to thank you again for your feedback and your continued support. Your suggestions significantly improved our work. We will include these additional comments and clarifications in the next version of the paper.

---

### Official Review · Reviewer_U2qE · 2023-07-10

**Soundness:** 3 good
**Presentation:** 2 fair
**Contribution:** 2 fair
**Rating:** 5
**Confidence:** 5

**Summary:**

The paper provides a generalization bound for a class of neural networks that consist of sparsely connected neurons. The class includes the class of convolutional neural networks and networks with weight sharing.
* The generalization bound (Theorem 3.2) is based on Rademacher complexity analysis (Proposition 3.1) that builds mainly on the work of Golowich et al (“Size independent sample complexity…”) and Peeling lemma (Lemma 3.4).  The Peeling lemma from Golowich et al cannot be directly used as there is a sum inside the square root in (3).
* The bound has linear dependence on the number of layers and depends on the product of the Frobenius norms of the kernel matrices (with the maximum taken per layer).
* The dimension dependence of the bound is via the notion of predecessor $\text{pred}(l,i)$, which capture how the neurons are connected across layers.

* As a conclusion, the authors provide a generalization bound for convolutional neural networks (Corollary 3.3).
* The experiments are done for MNIST and CIFAR-5m datasets, and for 4-layer convolutional networks. The main contribution of the paper is the generalization bound for sparse neural networks and its special case for convolutional networks.

To summarize, the paper considers an interesting setting, and the results provide interesting insights about the impact of neuron connectivity and channel dimensions. However, it seems to me that the authors can do a better job in supporting these claims experimentally, formulate their claims more precisely and report the limitations clearly.


**Strengths:**

The paper is in general well written (see my comments below). Obtaining non-vacuous generalization bounds for convolutional networks is an important and challenging topic. I find the formalization of DNNs as DAG is quite elegant and can model convolutional networks with their feature maps nicely. It is an interesting fact that the number of channels $H$ do not impact the bound (see Section 4.2 in the supplementary materials).

**Weaknesses:**

* Even if the predecessor set is minimally chosen to have the cardinality 2, the term $\prod_{l=1}^L |\text{pred}(l,j_{L-l})|=2^L$, which means exponential dependence on the number of layers. This issue is circumvented by assuming that the model is $\beta-$balanced (see Page 6 – line 211-212), and assuming $d_0 = 2^L$. This is a strong assumption, since  $d_0$, which is purely data dependent, becomes architecture dependent. It does not seem to me that this assumption is satisfied in practice, and is just used to remove the exponential dependence on the number of layers. The choice needs more justification.
* Note that in the plots, the term term $\prod_{l=1}^L |\text{pred}(l,j_{L-l})|$ is ignored, and only $\rho(w)/\sqrt{m}$ is plotted. Therefore, the claimed tightness and non-vacuity can be questionable.
* The proof seems to be a straightforward modification of the proof in Golowich et al, where only the Peeling lemma is updated. It seems to me that the technical novelty of the proof is limited, and the contribution lies mainly on the extension to the new case of sparse networks. This is fine, if the authors could provide better experiments. The considered architecture and datasets are simpler than what is typically tried in the generalization literature (see for example [44]; or in [17], AlexNet for cifar10 is considered). The message of the paper can become more compelling with better experiments.
* The plots only show the correlation between the generalization error and the bound. To show that the true generalization error does indeed scale with $\prod_{l=1}^L |\text{pred}(l,j_{L-l})|$, it is good to have additional experiments.
* I wonder about the message of Figure 2. The training and test errors saturate rapidly, and the plotted bound (red curve) does not seem to capture the correlation with the true generalization error (Figure 2  on supplementary materials seems to be even worse in terms of correlation). This might be because the true generalization error is already too small, and the authors might want to look for better visualizations.
* There are certain imprecise statements in the text that harm the overall presentation. For example, it is claimed that the bound is “quite tight”, although it is not clear how the tightness is measured and with respect to which baseline this judgment is formed. As another example, in the abstract it is claimed that “these bounds may be orders of magnitude better than standard norm based generalization bounds”, although no numerical comparison with many existing norm based bounds is done. Also the claim about the bound being non-vacuous is not always true based on the numerical results.  Admittedly, these limitations are addressed in the supplementary materials, Section 1. It might be a good idea to transfer this to the main part.


**Questions:**

* I am curious about the connection of this paper with two other works (see below). Arora et al also consider convolutional neural networks, and their analysis applies as well to sparse neural networks. Vardi et al prove a bound on a single hidden layer convolutional neural networks that is dependent on the spectral norm of the kernel, better than the bound in the paper, and the patch overlaps. They also need only Lipschitz  functions, which is also more general than the paper. I think it is useful to include this discussion and add these references to the main paper.

  * Arora, Sanjeev, Rong Ge, Behnam Neyshabur, and Yi Zhang. “Stronger Generalization Bounds for Deep Nets via a Compression Approach.” In International Conference on Machine Learning, 254–63, 2018.
  * Vardi, Gal, Ohad Shamir, and Nati Srebro. “The Sample Complexity of One-Hidden-Layer Neural Networks.” Advances in Neural Information Processing Systems 35 (2022): 9139–50.

## Comments and suggestions:
* The paper assumes 1-Lipschitz, positive-homogeneous activation functions (Lemma 3.4). Please clarify this throughout the paper.
* Adding a visualization of predecessor sets can be useful.
* I feel that the results on approximation guarantees (page 2) are not related to the generalization error and just motivate the case for sparse networks. I suggest moving it to the introduction.
* In page 2, it is mentioned that the MSE loss is used for training the network to get tighter bounds. The authors justify this choice by claiming that the weight norms are higher for “the cross-entropy loss which implicitly maximizes the network’s weight norms once the network perfectly fits the training data.” It would be good to add a reference for this, for instance:
** Daniel Soudry  et al, The Implicit Bias of Gradient Descent on Separable Data, JMLR 2018
* Lemma 2.2, which is Lemma 5.1, is quite standard. It would be enough to mention it in the supplementary materials. The proof is also quite standard and can be removed in my opinion.



**Limitations:**


The authors addressed some of the limitations in the supplementary materials, although this can be done more clearly.

---

> ### Author Rebuttal · Authors · 2023-08-08
>
> > Reviewer: “Even if the predecessor set is minimally chosen… the model is $\beta$-balanced (see Page 6 – line 211-212), and assuming $d_0=2^L$. This is a strong assumption, since $d_0$, which is purely data dependent…”
>
> Answer: Just to clarify, it is not the model that is assumed to be balanced but the data is balanced. The choice of $d\_\{0}=2^L$ is purely technical. This example would work perfectly with any other choice of $d_0$ and kernel size $k$. For example, we can still take $L=\lceil log\_k(d\_0) \rceil$ and kernel sizes $k$ with the last layer being fully connected. Then $\prod^{L}\_\{l=1} k\_l =k^{L-1}$, each term $z_{j}$ will be a patch of $k$ pixels. There are $\lfloor d\_\{0}/k \rfloor = k^{L-1}$ such patches. Therefore, we have $\max_\{j}\sum_\{i}||z\_\{j}(x_i)||^2 \cdot  k^{L-1} \leq \beta \cdot \sum\_\{i}||x_i||^2$ assuming the patches are $\beta$-balanced.
>
> > Reviewer: “Note that in the plots, the term $\prod^{L}\_\{l=1} pred(l,j\_\{L-l})$ is ignored, and only $\rho/\sqrt{m}$ is plotted…”
>
> Answer: We plotted the full bound as it is described in Theorem 3.2, including the above term (see line 351).
>
> > Reviewer: “The proof seems to be a straightforward modification… technical novelty of the proof is limited…”
>
> Answer: Our contribution is more conceptual than technical. 1. We demonstrate that the compelling derivation of Golowich can be expanded to convnets in a simple way, by incorporating sparsity. 2. This reveals a relationship between sparsity and generalization, that we believe could be significant for learning theory. 3. Through our experiments, we reveal that the previously deemed loose bounds are in fact tighter than anticipated. These bounds were originally evaluated for models trained with cross-entropy, which is a case where the norms of the weight matrices tend to grow. However, when training with MSE, the bounds are actually reasonable (in the sparse, i.e. convolutional, case).
>
> > Reviewer: This is fine, if the authors could provide better experiments... see for example [44]; or in [17], AlexNet for cifar10 is considered…
>
> Answer: It is difficult to experiment with off-the-shelf architectures (AlexNet) mostly because they incorporate layers that are difficult to handle with theory (e.g., pooling, dropout). Therefore, we took a vanilla architecture that we could easily analyze. It is worth noting that we still experimented with architectures of different sizes and depths and with CIFAR10 as in [17]. The goal of the experiments is to show that the bounds start to be reasonable in simple settings.
>
> > Reviewer: The plots only show the correlation between the generalization error and the bound... scale with $\prod^{L}\_\{l=1} pred(l,j_\{L-l})$...
>
> Answer: As mentioned above, throughout the experiments we calculated the full bound in Theorem 3.2. Furthermore, the goal of this paper is not to introduce the bound as a metric that can faithfully predict generalization. After all, there are much better ways to measure generalization than to derive generalization bounds. The goal of introducing this bound is to learn something about the relationship between sparsity and generalization, which is reflected in this bound.
>
> > Reviewer: I wonder about the message of Figure 2… This might be because the true generalization error is already too small…
>
> Answer: Indeed, the generalization gap appears fluctuating when plotted on logarithmic scales, primarily due to its comparatively smaller size relative to the other terms. However, it is actually quite flat and correlated with the other lines.
>
> > Reviewer: There are certain imprecise statements in the text… although no numerical comparison with many existing norm based bounds is done.
>
> Answer: When mentioning that our bounds are orders of magnitude smaller than existing ones, we mainly referred to bounds based on VC dimension and to the classic bound of Golowich et al. 2018, Long & Sedghi 2020. Following the reviews we conducted multiple experiments to verify that our bound is much smaller in practice compared to multiple bounds from the literature (see the comments for all of the reviewers and the attached pdf file). We will tone down these statements, and make them more specific and supported by the new experiments.
>
> The VC and the Long & Sedghi 2020 bounds scale with the number of trainable parameters, which make the bounds very loose for realistic networks (see lines 238-245). As we discussed in lines 62-70, a naive application of the Golowich et al. 2018 bound to Convolutional neural networks gives additional undesired factors to the bound that we were able to avoid. We will make it clearer in the next version of the paper.
>
> > Reviewer: Also the claim about the bound being non-vacuous… Admittedly, these limitations are addressed in the supplementary materials, Section 1. It might be a good idea to transfer this to the main part.
>
> Answer: We are happy to make it more explicit in the main text.
>
> > Reviewer: I am curious about the connection of this paper with two other works (Arora, Vardi)...
>
> Answer: We are happy to add these references along with discussions of these papers. Arora et al.’s paper presents a different type of bounds that apply to compressible networks. In essence, if a neural network is compressible then it is expected to have better generalization guarantees. While this work is relevant in fashion, it does not directly provide a fundamental distinction between fully-connected and convolutional networks. Looking at their bound in Tab. 1, it seems to be much looser than ours in practice. Regarding Vardi et al.'s paper, their findings seem to be similar in fashion to ours. However, as pointed out, their bound is only applicable to single-layer networks, which is considerably limiting.
>
> > Reviewer: comments and suggestions.
>
> Answer: Thank you very much! We will incorporate all of the suggestions (e.g., visualization of processor sets, moving text, citing Soudry, etc').

---

### Official Review · Reviewer_oZvZ · 2023-07-10

**Soundness:** 3 good
**Presentation:** 4 excellent
**Contribution:** 3 good
**Rating:** 6
**Confidence:** 4

**Summary:**

This paper presents a generalization bound for sparse neural networks, such as convolutional networks. The bound is based on a Rademacher complexity bound derived with a layer-peeling argument. Empirically, the bound is shown to be non-vacuous for a 3 layer network trained on MNIST.

**Strengths:**

This is a good paper with a strong contribution. As far as I know, it presents the first generalization bound based on Rademacher complexities that is non-vacuous on MNIST! However, I have some doubts (see weaknesses).

The paper is well written and easy to follow. The results are put into context and compared with prior work [1,2,3]. (For references see end of § Questions)

**Weaknesses:**

As already mentioned, I have doubt if the bound is really vacuous.
- The bound is evaluated on MNIST, i.e. a multi-class dataset. However, it seems that the bound only applies to binary classification (see Theorem 3.2). Extending the bound to multi-class settings can be done, for example with results from [5], but this yields an additional factor 10 = number of classes.
- In the provided source code, the maximum pixel sum is computed on the raw data, whereas training occurs on a normalized version of the training data from the data loader. This omits a factor $1/\sqrt{0.13} \approx 2.7$.
- Overall, this yields a factor of $\approx 27$ which results in a (comparably small but) vacuous bounds again.
- To clarify, even if the bound is vacuous, it still contributes a novel and significant contribution.

Comparisons with prior work is presented, but I have some additional questions, particularly regarding the scaling with network depth, which appears to be unfavorable in this work. See § Questions.

Throughout the paper it is claimed that the derived bounds are (or may be) *orders of magnitude* smaller than existing ones. However, this is not supported by empirical evidence, as existing bounds are not computed in the experiments. A table which lists numerical values of the bounds from [1,2,3,4] evaluated on the models from the MNIST experiment together with a short discussion would be insightful.

The bound is limited to homogeneous elementwise activation function (e.g. ReLU) and thus cannot handle some popular network components such as pooling layers. This is a weakness that is shared by all bounds proven with a layer-peeling argument, so this is only a minor concern, as long as it is clearly communicated and pointed out that existing bounds for conv nets are not limited in this regard (e.g. [2,3,4]).

**Questions:**

**Interpretation of the bound.**

Could you add a short paragraph after Proposition 3.1 which describes the different factors in the bound. Specifically, how should the maximum term inside the square root be interpreted? And, how does $z^0_{j_L}(x_i)$ differ from $x_i$?

Minor question: Its not entirely clear how to read the product. I guess it's $\big( \prod_l |\mathrm{pred}(l,j_{L-1})| \big) \sum_i z(x_i)$ and not $\big( \prod_l |\mathrm{pred}(l,j_{L-1})|  \sum_i z(x_i) \big)$

**Comparison with prior work \& scaling with depth.**

The paragraph starting at line 224 presents a comparison with [1] and shows that under specific architecture constraints, your bound is superior compared to [1] when applied to
conv nets. What about the other direction? When applying both bounds to an MLP, will both bounds be equal? Is yours still better or is it even worse?
At first sight, it seems that your bound has a malign exponential scaling with the network depth via $\prod_{l=1}^L |\mathrm{pred}(l,j_{L-1})|$ and will be worse for deep networks.

How does this exponential scaling affect the comparison with [3]? You state that your bound has a "*significantly better*" (explicit) dependence on depth (cf. l.253).
I also want to remark that one of the strengths of [3] is its benign scaling with the number of classes, which, as far as I see, Corollary 3 does not possess.

I would like to see a comparison with the generalization bound (♣) for conv nets from [4]. Both seem to scale similarly with a product of norms and a log term that is related to the number of parameters.
Moreover, the capacity reduction experiments in this reference are quite similar to your training procedure with a decomposition into weight direction and magnitude, only that lipschitz norms are controlled instead. Yet empirically the computed bound values in [4] are far from vacuous. I wonder why. Is it due to a strict better scaling of your bound? Or is it due to the training procedure, e.g., the network architecture or the mse loss?

You state that your bound and [3] "*cannot be directly compared, with each being better in different cases*". Could you specify such a setting where your bound is superior to [3] and [4]? As I already mentioned, I would like to see numerical values for the bounds from [1,2,3,4] in the MNIST experiment. Presumably, the MNIST experiment is already a setting where your bound is superior, but empirical evidence is needed.

**Notation.**
I did not understand the notation at the top of page 8 and in the proof of Proposition 5.3. What is $w^L \cdot \sigma (z^{L-1}_1(x), z^{L-1}_1(x))$ Is it
$w^ \cdot \sigma (z^{L-1}_1(x)) + w^L \cdot \sigma (z^{L-1}_2(x))$?

**Typo.**
"om" in line 145.

**References.**

[1] Golowich et al., Size-independent sample complexity of neural networks, COLT 2018
[2] Long & Sedghi, Generalization bounds for deep convolutional neural networks, IMCL 2020
[3] Ledent et al., Norm-based generalisation bounds for deep multi-class convolutional neural networks. AAAI 2021
[4] Graf et al., On Measuring Excess Capacity in Neural Networks, NeurIPS 2021
[5] Maurer,  A vector-contraction inequality for Rademacher complexities, ALT 2016


**Limitations:**

There is a section called "limitations" in the supplementary material, but therein, limitations are hardly discussed, only that the proposed bound "is generally loose and may be vacuous for certain settings". I think the opposite would have been more honest, i.e., that the bound is non-vacuous for certain settings but should be expected to be vacuous in general.

A limitation of the bound is that it only holds for certain activation functions and thus is not applicable to many network architectures, e.g. AlexNet. This is not sufficiently emphasized in the paper.

---

> ### Author Rebuttal · Authors · 2023-08-08
>
> > Reviewer: It seems that the bound only applies to binary classification…
>
> Answer: It’s possible to extend the bound to multiclass with only a logarithmic dependence on the number of classes. We will add a proof in the next version of the paper and we carefully reproduced the experiments with the new factors in the bound.
>
> The idea: We combine Lemma 3.1 in [1] with Corollary 4 in [2] with the multiclass margin loss from [1]. Finally, we only need to bound: $\mathbb{E}\_\{\epsilon} [\sup\_\{f \in \mathcal{F}}\sum\_\{i,j} \epsilon\_\{i,j} f\_\{j}(x_i)]$, where $\mathcal{F}$ is the set of neural networks of the form $Wg(x)$, where $g \in \mathcal{G}$ is a convolutional network as in the paper (with one output neuron with $c$ channels) and $W:\mathbb{R}^{c} \to \mathbb{R}^{k}$ is a matrix of norm at most $\alpha$. We can now use the top of page 31 in [3] to get a bound that scales with $\log(k)$ instead of $k$. This process eventually leads to replacing $1+\sqrt{2L\log(2deg(G))}$ with $2\sqrt{2}(1+\sqrt{2L\log(2deg(G)) + \log(k)})$ in our bound.
>
> [1] Bartlett et al., Spectrally-normalized margin bounds for neural networks, NeurIPS 2017.
>
> [2] Maurer, A vector-contraction inequality for Rademacher complexities, ALT 2016.
>
> [3] T. Galanti et al., Generalization Bounds for Few-Shot Transfer Learning with Pretrained Classifiers, 2023.
>
> > Reviewer: In the provided source code… whereas training occurs on a normalized version…
>
> Answer: We apologize for this honest mistake. We updated the code to take the normalization into account and reproduced our results.
>
> > Reviewer: ...numerical values of the bounds from [1,2,3,4] evaluated on the models from the MNIST experiment together with a short discussion would be insightful.
>
> Answer: Following the reviews, we conducted multiple experiments to compare our bound with the bounds in [1,2,3,4]. Please refer to the comment to all of the reviewers for details and the pdf for the plots. According to these experiments, our bound is smaller than the bounds in [1,2,3,4].
>
> As for the magnitudes, we referred to the VC dimension bounds, Golowich et al. 2018 and Long et al. 2020. The VC and the Long et al. 2020 bounds scale with the number of trainable parameters, which make the bounds very loose for realistic networks (see lines 238-245) . A naive application of the Golowich et al. 2018 bound to convnets gives additional factors that we were able to avoid (lines 62-70). We will make these statements more specific and supported by the new experiments in the next version of the paper.
>
> > Reviewer: Could you add a short paragraph after Proposition 3.1… different factors in the bound?
>
> Answer: We will add more information about each term in the bound and about calculating the maximum.
>
> > Reviewer: Its not entirely clear how to read the product…
>
> Answer: It is indeed the former expression, we will add brackets.
>
> > Reviewer: The paragraph starting at line 224 presents a comparison with [1] … When applying both bounds to an MLP, will both bounds be equal?
>
> Answer: The bounds are equal for MLPs. We can think of a fully-connected layer from $R^{n}$ to $R^{m}$ as a conv layer with kernel size 1x1, stride 1, $n$ input channels and $m$ output channels. The inputs and outputs of the layer are tensors of shapes $n\times 1 \times 1$ and $m \times 1 \times 1$. In our formulation, this means that the size of the predecessor set of the single neuron is 1 and the norm of the weight matrix is the norm of the linear transformation. Therefore, the bound conforms with the bound in [1].
>
> > Reviewer: “How does this exponential scaling affect the comparison with [3]? You state that your bound has a "significantly better" (explicit) dependence on depth (cf. l.253)... benign scaling with the number of classes…”
>
> Answer: We agree that our bound’s dependence on depth is not completely clear (e.g., it incorporates $\rho$ which implicitly depends on depth). We would like to mention that this is not the main point of the paper, as the goal of this work is to demonstrate the role of sparsity in generalization. We are happy to remove this claim. Regarding the scaling with the number of classes - as discussed above, we will extend our analysis to the multiclass setting and prove that the dependence on the number of classes is logarithmic (see above).
>
> > Reviewer: ...comparison with the generalization bound (♣) for conv nets from [4]... Is it due to a strict better scaling of your bound? Or is it due to the training procedure…?
>
> Answer: In [4] they trained the models with CE which tends to increase the norms of the weights of the network and eventually leads to large bounds. Following the reviews, we compared our bound with theirs in our setting (see attached pdf) and observed that their bound is still loose. Partially, because of their constants: when combining their Rademacher bound (in their page 18) and their Lemma 3.1 we obtain a generalization bound $8/n + 24\log(n)/\sqrt{n} \cdot \sqrt{\log(W)} \cdot (\sum^{L}\_\{i=1} (4 (\sup ||x||/\gamma) \cdot \prod\_\{l\neq i} s\_l \cdot ||K_\{i}||\_\{2,1})^{2/3})^{3/2}$. For example, their bound is at least 150 even if $\gamma=1$ and for each $i$, $ \sup ||x|| \cdot \prod\_\{l\neq i} s\_\{l} \cdot ||K\_\{i}||_{2,1}$ is equal to 1 (which is typically larger), and $W=500,000$, $n=50,000$, $L=5$.
>
> > Reviewer: I did not understand the notation at the top of page 8 and in the proof of Proposition 5.3.
>
> Answer: It is computed as follows: $w^L \cdot \sigma(z^{L-1}_1,z^{L-1}_2) = w^L_1\sigma(z^{L-1}_1)+w^L_2 \sigma(z^{L-1}_2)$, where $w^L_i$ is the $i$th component of $w^L$.
>
> > Reviewer: There is a section called "limitations"…  the bound is non-vacuous for certain settings but should be expected to be vacuous in general… only holds for certain activation functions
>
> Answer: We agree and we will make it more explicit in the next version of the paper. We will highlight the limitation to ReLU networks.

---

> > ### Comment · Reviewer_oZvZ · 2023-08-11
> >
> > Thank you for the detailed response. Most of my points were sufficiently addressed, but I have questions regarding the comparison with prior works ([3] and [4], see references in the initial review) and the numerical evaluation of your bound for convolutional networks.
> >
> > Surprised by the the large difference  (shown in Fig. 1 in the additional pdf) of a factor $2^{16}\approx 65k$ between your bound and the one from [4] and of $2^{22}\approx 4m$ to [3], I took a second look at the source code. Correct me if I am wrong, but **it seems that there is another, more severe, mistake when evaluating the bounds**, concerning the quantities $k_l$.
> >
> > The quantities $k_l$ are defined (cf. l217) as the input dimension of each neuron. From my understanding, this is the squared kernel size (because 2d convolution) times the number of channels of the preceding layer (so that when considering a fully connected layer as a conv layer with kernel size 1, $k_l$ is just the width). Yet, in the source code, $k_l$ is only the squared kernel size (see `utils.total_bound l.294`). Consequently, for constant output channels $H$, a factor of $H^{L/2}$ is missing, e.g. for the CONV-4-50, a factor of $50^2= 2.5k\approx 2^{11}$ and for CONV-4-500 a factor of $500^2=250k \approx 2^{18}$.
> >
> > ---
> > This potentially erroneous evaluation could clarify why your bound from Corollary 3.3 appears to be numerically much smaller than the related ones from the references [3] and [4] (see initial review). From a inspection of the quantities that appear in the bound, I could not explain this difference.
> >
> > In the following comparison I will focus on [4], as this bound (for me) is more easy to interpret, but presumably [3] behaves similar.
> > It seems, that both bounds (yours and [4]) have similar terms. Please clarify, if I misunderstood something.
> > - Numerical constants (1 in yours vs. 48 in [4])
> > - A product of weight norms (Frobenius vs Lipschitz)
> > - An average data norm ($\max_j \sum_i \Vert z_j^0(x_i) \Vert / \sqrt{m}$ vs $\sum_i \Vert x_i \Vert / \sqrt{m}$).
> >  - A logarithmic term corresponding to the number of parameters ($\sqrt{\log{2\mathrm{Deg}(G)}}$ vs $\sqrt{\log{2W}}$
> >
> > There is one difference though. Your bound has an additional factor of $\sqrt{\prod_l k_l} \approx \sqrt{k^L}$ whereas [4] contains a sum over (2,1)-distances divided by spectral norm, i.e., yielding a factor $\approx L  \sqrt{\text{kernelsize}^2 \cdot \text{out channels}}$.
> > Depending on the architecture the one or the other term might be favorable.
> > Long story short, it is unclear to me, what accounts for the large numerical difference by a factor of $\approx 2^{16}\approx 65k$ shown in Figure 1 of the pdf.

---

> > > ### Author Response · Authors · 2023-08-12
> > >
> > > We thank the reviewer for the additional comments.
> > >
> > > In Sec. 2.2, $k_l$ is $(\textnormal{kernel size})^2$ and not $(\textnormal{kernel size})^2 \cdot \textnormal{number input channels}$. For example, in line 161, a neuron $z^l\_{i}$ is computed by taking the concatenation $v^{l-1}_i$ of $|pred(l,i)|$ neurons $z^{l-1}\_{j}$, each of dimension $c\_{l-1}$ ($|pred(l,i)|=k_l$ for conv layers - line 174). We acknowledge that line 217 is a bit confusing as it gives the wrong impression that $k\_{l}$ is $(\textnormal{kernel size})^2 \cdot \textnormal{num input channels}$. We will make it clearer in the next version.
> > >
> > > As described in line 157, a network is a multi-layered graph, where each node $z^l_i$ (referred as a neuron) computes a vector of dimension $c_l$ (even though traditionally we think of a neuron as one variable). In this graph, each neuron $z^{l}\_{i}$ is connected to a set of predecessor neurons $pred(l,i)$ that holds $k\_{l}$ neurons from the previous layer.
> > >
> > > For a convolutional layer, each neuron is a vector of dimension $c_l$ (# output channels of the $l$th layer), $k_l$ represents the kernel size$^2$, and $pred(l,i)$ refers to the set of $k_l$ neurons (vectors of dimensions $c_{l-1}$) that are used to compute the $i$th neuron in the $l$th layer.
> > >
> > > For example, consider a convolutional layer that takes 3x64x64 images as input, it has 2x2 kernels, stride 2, and 300 output channels. Then, there are $32^2$ output neurons to this layer, each one of dimension 300, and each takes $k=4$ neurons as input. In our setting, $k_l$ is the kernel size squared of the $l$th layer. Meaning that for the 2x2 convolutional layers in the networks we used in the MNIST experiments, $k\_{l}=4$.
> > >
> > > We can represent a fully-connected (FC) network as follows. Suppose we have a FC network $W^L \sigma(W^{L-1} \dots \sigma(W^1x)\dots)$ of depth $L$ with dimensions $c_l$ in each layer. To represent this network, we can think of it as a graph with nodes $\cup^{L}\_{l=1} \\{z^l_1\\}$ with -one- neuron of dimension $c_l$ in the $l$th layer. The predecessor set of $z^l\_{1}$ is $\\{z^{l-1}\_{1}\\}$ which is of size $k_l=1$. Following the notations in line 161: $w^l\_{1} = W^l$. This gives an implementation of the FC network within our framework. Note that when representing FC networks this way, the bound in Corollary 3.3 conforms with the bound of Golowich et al. 2018.
> > >
> > > Next, we will estimate the ratio between the bound of [4] and our bound with CONV-3-500 trained on MNIST (learning rate $0.01$ and $\lambda=3e-3$). This estimation follows the high-level simplified analysis of the reviewer. The calculation we show demonstrate the huge gap between the two bounds, but when we directly calculate the bounds we get an even larger gap as shown in our plots.
> > >
> > > 1. Numerical constants (1 in ours vs. 48 in [4]).
> > >
> > > 2. When comparing the norm products (Frobenius in our bound vs. Lipschitz in [4]), their norm product is $\approx 0.18$ times ours.
> > >
> > > 3. We believe that in practice the terms $\sqrt{\max\_{j} \sum\_{i} ||z\_{j}(x\_{i})||^2}$ and $\sqrt{\sum\_{i} ||x\_{i}||^2}$ are not comparable. In practice, $||z\_{j}(x\_{i})||^2$ are much smaller than $||x_i||^2$ because $||x\_{i}||^2 = \sum\_{j} ||z^0\_{j}(x\_{i})||^2 = d_0 \cdot Avg\_{j} ||z^0\_{j}(x_i)||^2$. For a neural network with non-overlapping predecessor sets in each layer, $\prod^{L}\_{l=1}k_l \leq d_0$ (which is the case of CONV-3-500). Therefore, we have, $\sqrt{\sum\_{i} ||x\_{i}||^2} \geq \sqrt{\prod^{L}\_{l=1}k\_{l} \cdot \sum_{i} Avg\_{j} ||z^0\_{j}(x_i)||^2}$. Therefore, it makes more sense to compare between $\sqrt{\prod^{L}\_{l=1}k_l \max\_{j} \sum\_{i} ||z\_{j}(x\_{i})||^2}$ and $\sqrt{\sum\_{i} ||x\_{i}||^2}$ if the norms $||z^0\_{j}(x\_{i})||$ are fairly similar for all $i,j$. [Note, the same analysis can be made more generic for networks with onputs $c_0 \times d_0 \times d_0$ and different kernel sizes.]
> > >
> > > For the CONV-3-500 experiment on MNIST, the ratio between $\sqrt{\sum\_{i} ||x_i||^2}$ and $\sqrt{\prod^{L}_{l=1}k\_{l} \max\_{j} \sum\_{i} ||z\_{j}(x\_{i})||^2}$ is $\approx 2.5$.
> > >
> > >
> > > 4. We have a term $\sqrt{\log(2deg(G))}=\sqrt{\log(2\max\_{l} k_l)}$ vs. $\sqrt{\log(2W)}$ in [4]. For CONV-3-500 our term is $\log(2*4)\approx 2$ and theirs is $\log(2W)$ which is $\approx 14.5$ (since $W \geq 500^2 \cdot 4 = 10^6$ - the network contains one layer with 500 input an output channels with kernels of size $4=2^2$). Therefore, the ratio is $\sqrt{\log(2W)/\log(2deg(G))}\approx 2.7$.
> > >
> > >
> > > 5. Our bound includes a term $\sqrt{L}$ vs. [4] scales with $L$. The total number of layers in CONV-3-500 (including the convolutional and the FC layer - see lines 331-333) is $L=4$ and $L/\sqrt{L}=2$.
> > >
> > >
> > > 6. Their bound contains a factor of $\sqrt{\textnormal{kernel size}^2 \cdot \textnormal{output channels}}$. For CONV-3-500, it is $\sqrt{4\cdot 500}$.
> > >
> > > Taking into account all of the above ratios, we get: $48 \cdot 0.18 \cdot 2.5 \cdot 2.7 \cdot 2 \cdot \sqrt{4 \cdot 500} \approx 2^{12.5}$.

---

> > > > ### Comment · Reviewer_oZvZ · 2023-08-14
> > > >
> > > > Thank you for the additional clarifications. Indeed, there was a misunderstanding on my end of what is considered a neuron. I apologize for that.
> > > >
> > > > I took close look at the proofs of Lemma 5.2 and Proposition 5.3 and as far as I can tell, there is no error, i.e. the number of channels does not need to appear in the product under the square root.
> > > > I have to admit however, that I got confused by the notation and neuron picture, and that essentially I had to redo the proof by transforming everything to matrices $\in \mathbb R^{d_l \times d_{l-1}}$. In retrospective, I think my largest problem was to identify the dimensionality of the involved quantities, e.g. $W_j, w_j, w_j^l, v_j^l, \dots$ and I encourage the authors to specify, in a revised version, the corresponding spaces whenever a new variable is introduced, or it hasn't been used for a while.
> > > >
> > > > ---
> > > > Thank you also for quantifying my oversimplified comparison. A ratio of $2^{12.5} \approx 5.8k$ appears more realistic, particularly as the difference in numerical constants is already 48 and (presumably) the distance to initialization was not directly controlled during training.
> > > >
> > > > One question concerning point 3:
> > > > I agree that it is more appropriate to compare $\sum_i \Vert x_i \Vert^2$ to $\prod_l k_l \max_j \sum_i \Vert z_j(x_i) \Vert^2$ as I now understand that the product accounts for how often each pixel (stacked over channels) is used as input to the network.
> > > > If each pixel is used exactly once, then $\prod_l k_l = d_0$, and if predecessor sets overlap, then $\prod_l k_l > d_0$. The case $\prod_l k_l \gg d_0$ means that many input pixels are irrelevant for the models output, so it seems odd that the computed ratio is $2.5$.
> > > > I suspect that this is because the output of the last conv layer has more than one pixel, which are then flattened. But it seems that the flattening layer is not accounted for when computing the bound in the source code.
> > > > In other words, the last conv layer has multiple neurons, and so the neuron of the first fc layer has more than one predecessor.
> > > > Thus, as the third conv layer of CONV-3-H returns images of shape $H\times 3 \times 3$, we have $\prod_{l=1}^4 k_l = (2^3 * 3)^2 = 24^2$, and therefore I would have expected a ratio of only $7/8$.
> > > >
> > > > ---
> > > > I also wonder where the large difference of $\approx 2^{40} \approx 10^{12}$ to Golowich et al. comes from in Figure 1 (pdf). As discussed in the paragraph starting at line 224, the main difference is due to $\tilde \rho = \rho \sqrt{\prod_{l=1}^L d_l}$ (see l.187). So if there are 3 conv layers of stride 2 and the input is of shape 28 x 28, then only $\tilde \rho = 14 \cdot 7 \cdot 3 \rho = 294 \rho$

---

> > > > > ### Author Response · Authors · 2023-08-14
> > > > > **Thank you for helping us improve our paper**
> > > > >
> > > > > We really thank you for your dedicated attention and effort in improving our paper.
> > > > >
> > > > > > Reviewer: I encourage the authors to specify, in a revised version, the corresponding spaces whenever a new variable is introduced, or it hasn't been used for a while.
> > > > >
> > > > > Answer: We will make it clearer and more explicit in the next version of the paper.
> > > > >
> > > > > > Reviewer: One question concerning point 3
> > > > >
> > > > > Answer: We completely agree with the reviewer. Since the first fully-connected layer depends on multiple neurons (in this case 9=3^2 neurons), we will update the code to take into account the number of neurons that are used for this layer. Taking into account the number of neurons of the first fully-connected layer would give a different ratio than 2.5 (it should be 2.5/3=5/6). This changes the gap between our bound and the prior bounds, but our bound should still be tighter.
> > > > >
> > > > > > Reviewer: I also wonder where the large difference to Golowich et al. comes from in Figure 1 (pdf).
> > > > >
> > > > > Answer: As for Golowich's bound, we now realize that we miscalculated their bound.
> > > > > We forgot to take the square root of $\prod^{L}\_{l=1} d_l$ and also multiplied by $28^2$ (the input's spatial dimension) and by the dimensions of the last fully-connected layer. Since the dimensions of the last fully-connected layers are (num_channels * 3 * 3) x 10, this gives a factor of (28 * 14 * 7 * 3)^2 * 200 * 3 * 3 * 10 (which is approximately 2^40) instead of 14 * 7 * 3.
> > > > >
> > > > > We apologize and will definitely fix the calculation of the bound and replot the results. We note that the correction will not change the fact that Golowich's bound is still much larger than our bound.

---

> > > > > > ### Comment · Reviewer_oZvZ · 2023-08-14
> > > > > >
> > > > > > Thank you for your fast response.
> > > > > > All my question have been answered and I will consider updating my score during the discussion period.

---

> > > > > > > ### Author Response · Authors · 2023-08-15
> > > > > > > **Thanks again!**
> > > > > > >
> > > > > > > Thank you for your insightful and careful comments. Your feedback really improved our work.

---

### Author Rebuttal · Authors · 2023-08-09

We would like to thank all of the reviewers for their valuable time and feedback. Following the reviews we conducted several experiments to address the comments about the empirical results. Please see the attached pdf file for the plots.

Throughout the experiments, we compared our bound with several prior bounds appearing in [1,2,3,4]. We calculated the bounds for multiclass classification with 10 classes (with the additional factor depending on the number of classes that we added following the comments of reviewer oZvZ) and with margin $\gamma=1$ across all of the experiments for simplicity. We note that in [2] they did not explicitly mention their coefficient $C>0$ in their bound. We tried to calculate $C$ but it was very difficult as it relies on applying a complicated analysis from [5] (where constants are also not given). To compare with [2] we simply assumed $C=1$, even though we expect it to be larger than 1.

[1] Golowich et al., Size-independent sample complexity of neural networks, COLT 2018

[2] Long & Sedghi, Generalization bounds for deep convolutional neural networks, ICML 2020

[3] Ledent et al., Norm-based generalization bounds for deep multi-class convolutional neural networks. AAAI 2021

[4] Graf et al., On Measuring Excess Capacity in Neural Networks, NeurIPS 2021

[5] Giné and Guillo, On consistency of kernel density estimators for randomly censored data: rates holding uniformly over adaptive intervals. In Annales de l’IHP Probabilités et statistiques, volume 37, pages 503–522, 2001.

Figure 1 (a): A convolutional network trained on MNIST. Following the comments of reviewers oZvZ, U2qE, dgpJ, we compare our bound with [1,2,3,4] when varying the width of the network. As can be seen, our bound is largely unaffected by increasing the width, while the bounds in [1,2,4] generally increase with the width (especially [1]). This is a very nice advantage of the bound because increasing the width grows the number of trainable parameters.

Figure 1 (b): Following the comments of all of the reviewers, we investigated the effect of depth on the various bounds. As can be seen, similar to [1,3,4], our bound seems to grow with the depth of the network. As mentioned in the responses to the reviewers, the focus of this paper is to extend the compelling derivation of Golowich to convolutional networks by incorporating sparsity, to reveal a relationship between sparsity and generalization, and to show that our bound is reasonable in practice. However, it would definitely be interesting to relax the dependence of this bound on depth as future work.

Figure 2: Following the suggestion of reviewer dgpJ, we studied the correlation between different bounds and the generalization gap when varying $\lambda$. As can be seen, both the generalization gap and our bound decrease when increasing $\lambda$.

Figure 3: For completeness, we also plotted the bounds during training when considering the setting of Figure 2 with $\lambda=3e-3$.

---

### Comment · Area_Chair_4tHP · 2023-08-11

Hi all,

Thanks for serving as the reviewers for this submission. As the authors have already turned in their responses. It is our turn to start the further discussion. Here is a to-do list:

(1) Please acknowledge the authors when you finish reading their responses.
(2) Please indicate whether you have any further questions for the authors such that they can continue to response.
(3) Please indicate whether you are willing to change the ratings.

Best

AC

---

### Decision · Program_Chairs · 2023-09-21

**Decision:**

Accept (poster)

**Comment:**

This paper presents an interesting theoretical analysis of Rademacher complexity bounds for deep sparse neural networks. The reviewers agree that the theoretical contributions are solid, providing new bounds that leverage network sparsity and have the potential to improve upon previous results.

However, there are concerns about the empirical evaluation of the bounds and some of the resulting conclusions. In particular, there appear to be errors in numerically estimating the bounds that affect comparisons to prior work. This calls into question statements about the tightness of the bounds and their ability to be non-vacuous in practice.

I believe the core theoretical results are sound and provide value to the community's understanding of generalization bounds for sparse networks. However, the authors need to address the issues raised regarding the experiments and revise any overstated conclusions based on those results. Given the high quality of the theory, I recommend conditionally accepting the paper pending revisions to the empirical methodology and tempering of the conclusions.

Specifically, the authors should carefully re-evaluate the bound computations and comparisons to prior works, address any errors, and revise the conclusions accordingly. Claims about the tightness of the bounds and their non-vacuousness need to be qualified based on the corrected calculations. The limitations discussed regarding uniform convergence bounds should also be acknowledged.

With these revisions, I believe the paper will make a solid contribution. I recommend acceptance pending modifications to the experiments and conclusions.